# No Loss, No Gain: Gated Refinement and Adaptive Compression for Prompt Optimization

**Wenhang Shi**[1]    **Yiren Chen**[2]    **Shuqing Bian**[3]    **Xinyi Zhang**[1]    **Kai Tang**[3]
**Pengfei Hu**[3]    **Zhe Zhao**[3]    **Wei Lu**[1]    **Xiaoyong Du**[1]*

[1]Renmin University of China
[2]Peking University
[3]Tencent

{wenhangshi, xinyizhang.info, lu-wei, duyong}@ruc.edu.cn,
yrchen92@pku.edu.cn, shuqingbian@gmail.com,
{aydentang, alanpfhu, nlpzhezhao}@tencent.com

## Abstract

Prompt engineering is crucial for leveraging the full potential of large language models (LLMs). While automatic prompt optimization offers a scalable alternative to costly manual design, generating effective prompts remains challenging. Existing methods often struggle to stably generate improved prompts, leading to low efficiency, and overlook that prompt optimization easily gets trapped in local optima. Addressing this, we propose GRACE, a framework that integrates two synergistic strategies: **G**ated **R**efinement and **A**daptive **C**ompression, achieving **E**fficient prompt optimization. The gated refinement strategy introduces a feedback regulation gate and an update rejection gate, which refine update signals to produce stable and effective prompt improvements. When optimization stagnates, the adaptive compression strategy distills the prompt's core concepts, restructuring the optimization trace and opening new paths. By strategically introducing information loss through refinement and compression, GRACE delivers substantial gains in performance and efficiency. In extensive experiments on 11 tasks across three practical domains, including BIG-Bench Hard (BBH), domain-specific, and general NLP tasks, GRACE achieves significant average relative performance improvements of 4.7%, 4.4% and 2.7% over state-of-the-art methods, respectively. Further analysis shows that GRACE achieves these gains using only 25% of the prompt generation budget required by prior methods, highlighting its high optimization efficiency and low computational overhead. Our code is available at https://github.com/Eric8932/GRACE.

## 1   Introduction

Large language models (LLMs) exhibit impressive generalization abilities, being able to perform various tasks based on simple instructions [40, 10]. However, downstream tasks often impose specific requirements that require adaptations beyond these general capabilities. To bridge this gap, prompt engineering has emerged as a lightweight alternative to traditional fine-tuning, aiming to craft effective prompts that unlock the full potential of LLMs [20]. Some automatic methods adapt model training by fine-tuning soft prompts or using reinforcement learning to combine discrete tokens, but they rely heavily on access to the internal states or gradients of LLMs [14, 50]. For advanced API-based LLMs like GPT-4 [1], prompt engineering remains a complex and labor-intensive process, often requiring human experts with deep insight into both LLM behavior and task-specific nuances.

---

*Corresponding author

39th Conference on Neural Information Processing Systems (NeurIPS 2025).

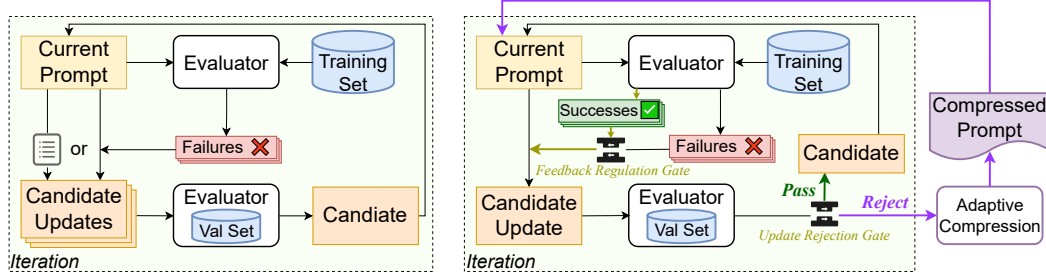

Figure 1: **Comparison of Prompt Optimization Methods.** (a) illustrates the traditional process of prompt expansion and selection. (b) presents our GRACE framework, implementing a two-stage gated refinement and adaptive compression to enable more effective and efficient optimization.

Recent methods automate prompt generation for closed-source LLMs by employing LLMs as optimizers to iteratively expand and select prompt candidates, as shown in Figure 1 (a) [51]. They can be broadly categorized into two lines based on their expansion strategies. One line of work generates candidates using heuristics such as text edits or paraphrasing [45, 24, 12, 11]. Such search-based methods lack clear optimization guidance, often producing prompts with random modifications that remain semantically close to the original. The other line of work leverages the reflection capabilities of LLMs, iteratively revising prompts based on analyses of failed training samples [25, 47, 48, 43, 37]. While error feedback provides strong update signals, they can be overly aggressive and biased without proper regulation, frequently causing prompt overcorrection and semantic drift. These unstable updates make it difficult to produce improved prompts. Consequently, existing methods typically generate a large number of candidates at each step to secure prompt improvement, leading to inefficient optimization and high computational costs [21]. Moreover, they often overlook that prompt optimization is prone to getting trapped in local optima, with performance plateauing after only a few update steps. Search-based methods often make minimal prompt changes, struggling to achieve continuous progress in the discrete prompt space. Reflection-based methods, though more aggressive, tend to incorporate increasingly instance-specific information as prompts are enriched, which lacks generalization and yields no performance gains.

To address the above limitations, we introduce GRACE, an efficient automatic prompt optimization framework, which integrates two synergistic strategies: gated refinement and adaptive compression. As shown in Figure 1 (b), GRACE iteratively controls prompt updates through gated refinement and escapes local optima via adaptive compression. At each iteration, a candidate update is generated under the guidance of a feedback regulation gate, which leverages successful training samples to regulate the update signals from failed ones. The candidate then goes through an update rejection gate, which blocks the information if it fails to improve validation performance. This two-stage gating mechanism refines the information flowing into each prompt update, enabling more stable and efficient improvement without excessive candidate generation. When repeated rejections occur, adaptive compression is triggered to remove redundant content and abstract overly specific details in the prompt. The compressed prompt restructures the optimization landscape and opens new directions for gated refinement, facilitating escape from local optima. Together, these two information-loss strategies form a synergistic loop, alternating between local refinement and global restructuring, which achieves a strong balance between exploration and exploitation in the vast prompt space.

**Contributions** (1) We propose GRACE, an efficient prompt optimization framework, which strategically introduces information loss to achieve stable and sustained prompt improvements and effectively escape local optima. (2) GRACE demonstrates strong generality across 11 tasks spanning three practical and distinct domains: BIG-Bench Hard (BBH) [35], domain-specific, and general NLP tasks, achieving average relative performance improvements of 4.7%, 4.4%, and 2.7% over state-of-the-art prompt optimization methods, respectively. (3) GRACE exhibits significantly higher optimization efficiency: whereas prior methods generally require generating over 300 prompts to converge, GRACE reaches superior final performance using fewer than 80 prompts, substantially reducing overhead.

# 2  Methodology

Given a base LLM $\mathcal{B}$ and a target task $\mathcal{T}$, the goal of automatic prompt optimization is to discover a natural language prompt $\mathcal{P}^{\mathcal{T}}$ that effectively bridges the gap between the general capabilities of $\mathcal{B}$ and the specific requirements of $\mathcal{T}$. Most existing methods leverage an auxiliary optimizer LLM $\mathcal{O}$ to iteratively sample local prompt variants or revise prompts based on model errors. However, they often underutilize task data to generate appropriate updates, leading to inefficient optimization and frequent convergence to local optima. To address this, we introduce GRACE, an efficient prompt optimization framework designed to produce effective prompts via gated refinement and adaptive compression strategies, striking a balanced exploration-exploitation dynamic in the vast prompt space.

**Problem Formulation** Following the standard prompt optimization setting [51, 37], we start with an initial prompt $\mathcal{P}_0$ and a small set of training and validation samples drawn from the target task dataset $D = (q_i, a_i)_{i=1}^N$, where each $(q_i, a_i)$ denotes a question-answer pair. Given the model input consisting of $\mathcal{P}$ and $q_i$, the base LLM $\mathcal{B}$ makes the prediction based on $p_{\mathcal{B}}(a_i \mid \mathcal{P}, q_i)$. The goal of prompt optimization is to find an optimal prompt $\mathcal{P}^*$ that maximizes the performance of $\mathcal{B}$ on task $\mathcal{T}$ towards a scoring function $f$ (e.g. accuracy). This can be formalized as an optimization problem:

$$\mathcal{P}^* = \underset{\mathcal{P} \in \mathcal{S}}{argmax}\ f_{\mathcal{B}}(\mathcal{P}, D) = \underset{\mathcal{P} \in \mathcal{S}}{argmax} \sum_{(a_i, q_i) \in D} f(p_{\mathcal{B}}(a_i \mid \mathcal{P}, q_i)), \tag{1}$$

where $\mathcal{S}$ denotes the prompt search space, an infinite and intractable space, if not impossible, to comprehensively enumerated. Next, we introduce GRACE framework and detail its core strategies.

## 2.1  GRACE Framework

GRACE efficiently updates prompts and overcomes frequent convergence to local optima by strategically incurring loss of redundant or detrimental information. It introduces two synergistic strategies, gated refinement and adaptive compression, and integrates them in an iterative process, as shown in Figure 1 (b). At each iteration, GRACE first applies gated refinement to update the current prompt. A candidate update is generated under the control of a feedback regulation gate, which leverages successful samples to refine error signals from failed samples. The update information then goes through an update rejection gate, which may apply further filtering. When repeated rejections occur, indicating optimization stagnation, GRACE activates adaptive compression to escape local optima by simplifying and abstracting the prompt. The pseudocode can be found in Appendix Algorithm 1.

The two strategies work in coordination to form a recurrent optimization loop executed over $T$ iterations. Initially, gated refinement enables fine-grained improvements to the prompt. When these incremental updates become ineffective, adaptive compression resets the optimization trajectory by distilling the prompt into a more general and compact form. This compressed prompt then serves as a new starting point for further gated refinement. By alternating between local refinement and global restructuring, GRACE performs both local exploitation and global exploration in the prompt space, enabling more efficient and stable optimization.

## 2.2  Gated Refinement: Two-Stage Information Filtering for Stable Updates

To ensure only beneficial information flows into prompt updates, GRACE employs a two-stage gated refinement strategy: (1) generating effective updates via a feedback regulation gate; (2) selectively adopting updates via an update rejection gate.

**Feedback Regulation Gate** The generation of candidate updates is guided by a feedback regulation gate that refines signals from failed training samples using successful ones. Failure feedback is widely used as the primary update signal, analogous to a gradient [25]. To prevent overly strong or biased failure signals from corrupting the prompt, GRACE incorporates feedback from successful samples as a regularization gate [22], leveraging their known effective patterns to control and balance the content and magnitude of updates. At

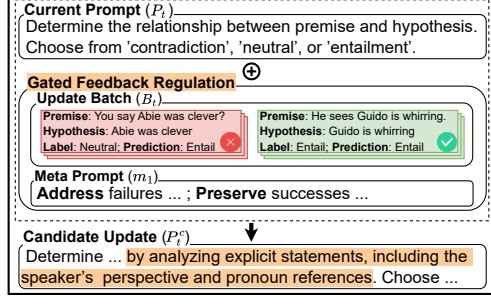

Figure 2: Update using feedback regulation gate.

each iteration $t$, GRACE categorizes the training samples $D_{\text{train}}$ into successes $S_t$ and failures $F_t$ based on performance of $\mathcal{B}$. As illustrated in Figure 2, it samples $S'_t \subseteq S_t$ and $F'_t \subseteq F_t$ to construct an update batch $B_t = S'_t \cup F'_t$, and generate a candidate update $\mathcal{P}^c_t$ using $\mathcal{O}$ as:

$$\mathcal{P}^c_t \sim p_{\mathcal{O}}(\mathcal{P} \mid \mathcal{P}_t, B_t, m_1), \tag{2}$$

where $m_1$ is a meta-prompt instructing $\mathcal{O}$ to revise $\mathcal{P}_t$ by addressing errors in $F'_t$ while preserving effective patterns in $S'_t$. This feedback regulation gate balances update signals by losing information, mitigating the risks of overfitting to failure cases and enhancing the stability of prompt improvements.

**Update Rejection Gate** Once a candidate update is generated, GRACE employs an update rejection gate to determine its adoption, further refining the information flow. Given the high prompt sensitivity of $\mathcal{B}$ [28], even updates from balanced signals may contain redundant or harmful information and can impair optimization. To avoid such degradation, GRACE evaluates the candidate update $\mathcal{P}^c_t$ using a validation set $D_{\text{val}}$ and scoring function $f_{\mathcal{B}}$. The updated prompt for the next iteration is chosen as:

$$\mathcal{P}_{t+1} = \underset{\mathcal{P} \in \{\mathcal{P}_t, \mathcal{P}^c_t\}}{argmax} f_{\mathcal{B}}(\mathcal{P}, D_{val}). \tag{3}$$

If the candidate fails to improve performance, it is rejected, meaning the gate blocks this update information. This update rejection gate discards unnecessary or detrimental updates, further ensuring that only meaningful and beneficial information is incorporated into the prompt updates.

Together, the two-stage gating mechanism introduces controlled information loss to enable more targeted and stable prompt updates, thereby enhancing optimization efficiency. By leveraging successful samples to regulate error signals, GRACE produces more effective updates, reducing the need for excessive candidate generation and evaluation. Moreover, the rejection gate further filters out potentially harmful information, mitigating unpredictable and abrupt changes in prompt behavior.

### 2.3 Adaptive Compression: Information Distillation for Escaping Local Optima

As prompt updates progressively enrich the prompt, the added information often shifts from generalizable guidance to increasingly case-specific and concrete details. This over-specification traps the optimization in local optima by overfitting to narrow patterns, leading to stagnation in performance improvement. To address this, GRACE introduces an adaptive compression strategy that activates when optimization stagnates. Specifically, when the rejection

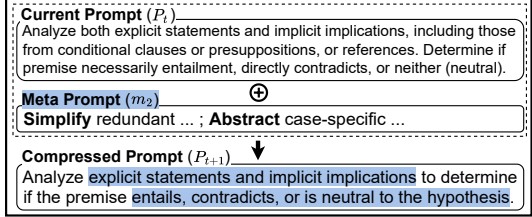

Figure 3: Update using adaptive compression

gate blocks $K$ consecutive update candidates, GRACE compresses current prompt $\mathcal{P}_t$ to distill its core concepts as shown in Figure 3. The compression is performed as:

$$\mathcal{P}_{t+1} \sim p_{\mathcal{O}}(\mathcal{P} \mid \mathcal{P}_t, m_2), \text{ when } \sum_{j=t-K+1}^{t} \mathbb{I}[\mathcal{P}_j = \mathcal{P}_{j-1}] = K. \tag{4}$$

Here, $m_2$ is a meta-prompt that instructs $\mathcal{O}$ to both simplify the current prompt by merging or removing redundant elements, and abstract away concrete, instance-specific instructions (e.g., narrow conditionals or memorized phrasings) into more broadly applicable guidance. By introducing information loss in the prompt, the adaptive compression not only helps escape from local optima, but also provides a better optimization starting point, opening up new directions for gated refinement.

The adaptive compression strategy inherently aligns with Information Bottleneck theory [29], which posits that an optimal representation should compress input data while preserving task-critical information. By removing redundant and overly specific content, GRACE emphasizes essential, task-relevant patterns and actively pursues this information bottleneck. Therefore, the information loss from compression enhances generalization and paves the way for more sustained optimization.

## 3 Experiments

**Tasks and Datasets** We conduct comprehensive experiments on 11 tasks across three distinct domains: 5 BIGBench Hard (BBH) taks [35], requiring complex reasoning or domain knowledge; 3 biomedical

|            | BBH   | Domain-Specific Tasks |         |       |       | General NLP Tasks |       |       |       |
|            | Avg.  | NCBI  | Biosses | MedQA | Avg.  | Subj  | TREC  | CB    | Avg.  |
|------------|-------|-------|---------|-------|-------|-------|-------|-------|-------|
| Task (ZS)  | 77.45 | 60.83 | 72.50   | 84.75 | 72.69 | 64.20 | 66.20 | 89.29 | 73.23 |
| Task (FS)  | 72.73 | 64.90 | 65.00   | 79.25 | 69.72 | 85.00 | 69.80 | 92.86 | 82.55 |
| CoT (ZS)   | 77.74 | 60.02 | 72.50   | 85.75 | 72.76 | 59.10 | 64.80 | 94.64 | 72.85 |
| CoT (FS)   | 79.62 | 64.69 | 67.50   | 84.50 | 72.23 | 84.00 | 73.40 | 94.64 | 84.01 |
| EvoPrompt  | 81.15 | 70.96 | 70.00   | 84.75 | 75.24 | 92.30 | 85.40 | 89.29 | 89.00 |
| OPRO       | 85.51 | 69.47 | 72.50   | 85.50 | 75.82 | 94.60 | 86.40 | 89.29 | 90.10 |
| APO        | 88.14 | **73.83** | 67.50 | 85.50 | 75.61 | 94.80 | 90.60 | 96.43 | 93.94 |
| PromptAgent| 89.42 | 71.81 | 75.00   | 86.00 | 77.60 | 91.50 | 90.20 | 94.64 | 92.11 |
| GRACE      | **94.13** | **73.83** | **85.00** | **86.50** | **82.00** | **95.70** | **94.20** | **100** | **96.63** |

Table 1: Performance on 3 types of tasks. Metrics are accuracy, except F1 for NCBI. ZS/FS denote Zero-Shot and Few-Shot settings. Task (ZS) is the initial prompt for prompt optimization methods. BBH Performance is averaged on five challenging tasks, and the bold values indicate the best.

domain-specific tasks, including NCBI [8], Biosses [31], and Med QA [15]; 3 general NLP tasks, including TREC [36], Subj [23], and CB [4]. Details of tasks and datasets are in Appendix A.1.

**Baselines** We compare GRACE against two categories of prompt baselines: (1) manually designed prompts and (2) automatic prompt optimization methods. For manual prompts, we include simple task-related instructions from the original datasets as Task (ZS), and a Chain-of-Thought prompt "Let's think step by step" as CoT (ZS) [17]. We also include the few-shot versions [41]: for BBH tasks, exemplars are sourced from [35], and for the remaining tasks, exemplars are constructed from the training data. For automatic prompt optimization, we compare against the following methods:

- **EvoPrompt** [12] iteratively generates candidate prompts using evolutionary algorithms, including genetic algorithms or differential evolution, representing search-based methods.
- **OPRO** [47] generates candidate prompts based on the history of previous prompts and their evaluation scores, and can be viewed as a hybrid of search-based and reflection-based methods.
- **APO** [25] uses the reflective capability of the optimizer LLM to generate text gradients from model errors, which are then used to revise the prompt, representative of reflection-based approaches.
- **PromptAgent** [37] formulates prompt optimization as a planning task and employs a Monte Carlo Tree Search framework. It relies on error feedback for iterative updates and is reflection-based.

**Implementation Details** Since prompt optimization requires complex reasoning over sample-level analysis and prompt update, we employ DeepSeek-R1 as the optimizer LLM [6]. The base LLM is DeepSeek-V3-0324 [5] . All optimization methods start with the same initial prompt, Task (ZS), except for EvoPrompt, which uses 14 additional variants. During optimization, all methods follow a similar procedure: candidate prompts are generated then evaluated on a held-out validation set (separate from training samples). Once optimization is complete, the prompts with the highest validation performance are evaluated in a test set (disjoint from the training and validation sets), and the best test result is reported. To ensure a fair comparison, we set the maximum number of generated prompts for all baseline methods to approximately 300, following prior work and ensuring general convergence [21, 43]. For GRACE, we set the maximum number of iterations to $T = 80$, which requires significantly fewer prompts to converge. Moreover, we sample 3 success and 3 failure examples ($|S_t'| = |F_t'| = 3$) to form the update batch, and trigger compression after $K = 5$ consecutive rejections. Detailed hyperparameter settings for all methods are in Appendix A.2.

## 3.1 Main Results

Table 1 presents a comprehensive comparison of the final prompts produced by GRACE against baseline methods across three task categories. On the BBH tasks, GRACE consistently outperforms all baselines, achieving average relative improvements of at least 14.5% over manual prompts and 4.7% over other optimization methods. The superior performance of optimization methods over well-crafted manual prompts like CoT (FS) underscores the effectiveness of leveraging LLMs for automatic prompt optimization. However, the gains from search-based methods are marginal, particularly for EvoPrompt. Lacking clear optimization direction, these methods often converge to prompt variants that remain semantically close to the initial prompt, resulting in only minor improvements. Reflection-

based methods, such as APO and PromptAgent, incorporate explicit error-driven signals to revise prompts. While these signals can be informative, they are often overly strong and biased when unconstrained, leading to excessive update magnitudes and a higher tendency to get trapped in local optima, ultimately limiting performance. In contrast, GRACE combines gated refinement, which dynamically adjusts update magnitude and content, with adaptive compression, which helps escape local optima. Together, these information-loss mechanisms enable GRACE to achieve superior final performance across diverse tasks. In Appendix Table 6, we report per-task BBH results and additional comparisons using GPT-4.1 [1] as the base LLM. Moreover, Appendix Table 7 presents transfer evaluations of prompts optimized on DeepSeek-V3, demonstrating the superior cross-model transferability of GRACE-optimized prompts.

On domain-specific and general NLP tasks, GRACE continues to demonstrate notable gains, achieving average relative improvements of 4.4% and 2.7% over state-of-the-art prompt optimization methods, respectively. These results indicate that GRACE is not only effective in challenging scenarios such as BBH, but is also capable of integrating domain-specific and general knowledge to craft high-quality, gap-bridging prompts across diverse tasks. Moreover, in Appendix Table 8, we further evaluate two summarization datasets, where GRACE again attains the best performance. This versatility highlights GRACE's broader applicability and robustness in real-world prompt engineering settings.

## 3.2 Convergence Curve Analysis

To better understand the reasons behind performance differences across methods, we further analyze their optimization processes. Figure 4 presents the convergence curves on the TREC task, plotting the test performance of the best-discovered prompt against the cumulative number of prompts generated. For baseline methods, each point represents an optimization step, while for GRACE, which generates one candidate per step, points are marked only when the update improves performance. We observe that baseline methods generally plateau after a few updates, indicating premature convergence to local optima. This issue is particularly evident in reflection-based methods, where most performance gains occur in the first 2–3 steps. Although error feedback provides effective update signals,

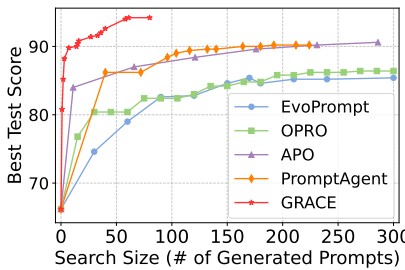

Figure 4: Changes of best test score as number of generated prompts increases.

it gradually introduces instance-specific content, the accumulation of which reduces prompt generalizability and yields no performance gains. Search-based methods exhibit more gradual improvement, but as optimization progresses, their limited updates become increasingly difficult to make meaningful gains in the discrete prompt space, resulting in lower final performance. In contrast, GRACE demonstrates a stable and sustained optimization trajectory, characterized by rapid initial gains followed by steady improvements, culminating in the best final performance.

Beyond final performance, efficiency is also critical in prompt optimization. GRACE consistently maintains higher performance under the same number of generated prompts, and reaches a better endpoint using significantly fewer prompts, underscoring its efficiency. By contrast, baseline methods not only tend to get stuck in local optima, but also exhibit slower ascent due to their inability to stably generate improved updates. Baseline methods either perform small, random updates or make overly aggressive changes, both of which lead to unstable prompt updates. This instability forces them to explore a large number of candidates at each step to find improved prompts, greatly reducing optimization efficiency. These limitations underscore the importance of strategies that enable appropriate adjustment of updates and facilitate escape from local optima in GRACE, which achieves a balanced and efficient exploration-exploitation dynamic in the prompt space.

## 3.3 Ablation Study: How Loss Leads to Gain

To investigate how GRACE transforms information loss into gains in performance and efficiency, we perform an ablation study on its core design components. Figure 5 compares the convergence curves of various GRACE configurations on the TREC task. It plots the validation performance of each candidate update and highlights points where adaptive compression is triggered, along with the final test score corresponding to the peak validation point.

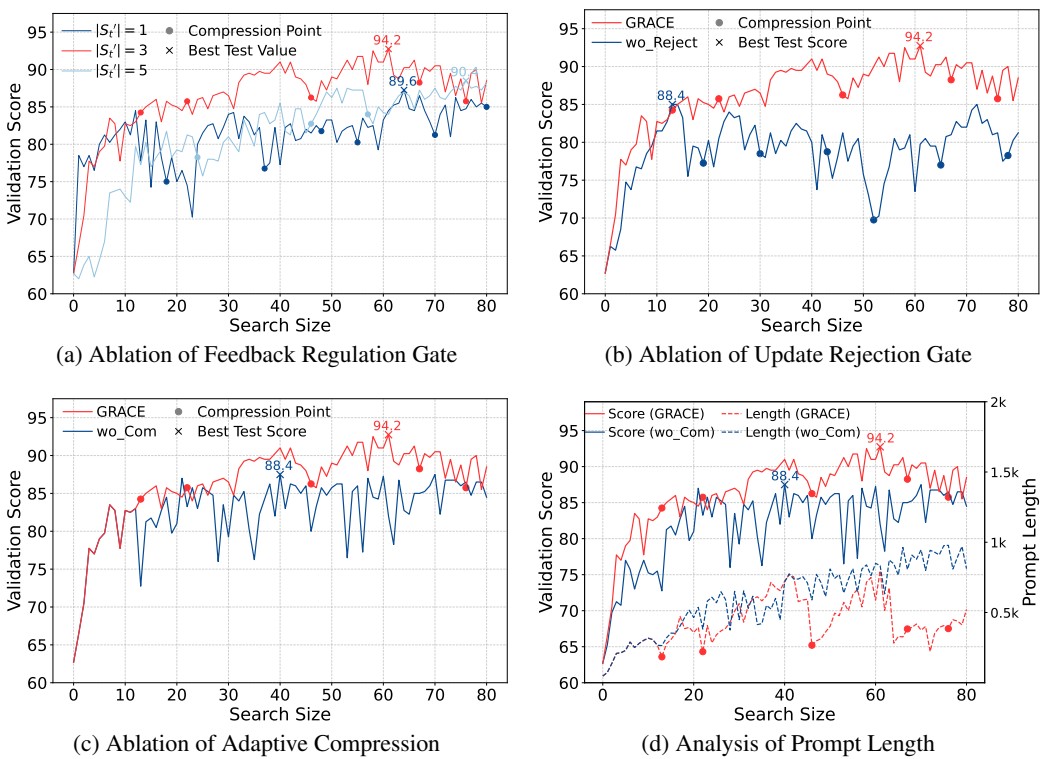

Figure 5: Ablation study on TREC task: (a) Effect of positive/negative sample ratio ($|S_t'|+|F_t'| = 6$); (b) Effect of accepting all candidate updates (wo_Reject); (c) Effect of removing adaptive compression (wo_Com); (d) Connection between performance and prompt length. Prompt is compressed in Compression Point and the test score corresponding to the best validation score is shown.

**Ablation on Feedback Regulation Gate** To examine the role of success samples in the feedback regulation gate, we vary the ratio of success to failure samples in the update batch, while keeping the total batch size fixed ($|B_t| = 6$). Figure 5a presents the results of different $|S_t'|$. A higher proportion of success samples leads to slower but more sustained performance improvements, confirming their regularization effect. However, imbalanced ratios yield suboptimal outcomes. When error signals are dominant due to insufficient regulation ($|S_t'| = 1$), performance improves mainly in the early stages. The compression merely results in repeated convergence to local optima, with limited further gains. Conversely, when the update signals are weak ($|S_t'| = 5$), the optimization trace is more stable. But in the discrete prompt space, overly conservative updates slow convergence and raise the risk of stagnation, limiting long-term gains. These findings highlight the importance of appropriate information loss in update signals, as both extremes hinder effective prompt improvement. Thus, a balanced feedback regulation mechanism is essential for achieving stable and efficient optimization.

**Ablation on Update Rejection Gate** To assess the impact of GRACE's update rejection gate, we conduct an ablation in Figure 5b, where prompts are updated with every generated candidate. For consistency, adaptive compression remains active and is triggered when performance fails to improve for $K$ consecutive steps. This greedy update strategy exhibits behavior similar to that observed under insufficient success regulation in Figure 5a, with the optimization rapidly and repeatedly converging to suboptimal local optima. While the feedback regulation gate helps control update signals, it can still introduce noisy, redundant, or even harmful information into the prompt, leading to further updates fail to bring meaningful gains. These results highlight the importance of GRACE's update rejection gate, which acts as a safeguard: by selectively incorporating only beneficial information, it helps maintain stable and sustained optimization.

**Ablation on Adaptive Compression** Prompt optimization is prone to getting trapped in local optima, a challenge that is often overlooked. To investigate whether adaptive compression effectively mitigates this issue, we conduct an ablation study in Figure 5c. Although GRACE's gating mechanism promotes

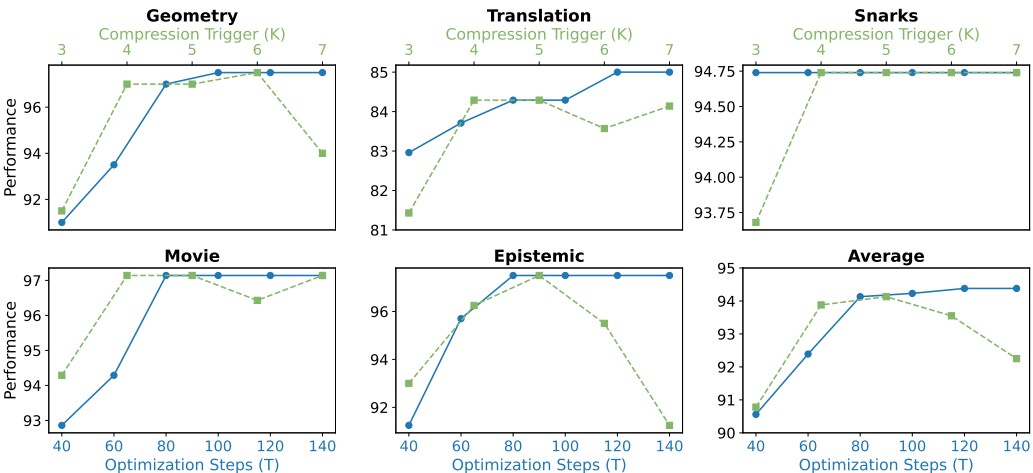

Figure 6: Ablation study for the optimization steps $T$ and the compression trigger $K$ on BBH tasks.

stable updates, removing compression leads to improvements mainly in the early stages, followed by stagnation and fluctuations around a local optimum. In contrast, when compression is triggered upon stagnation, this limitation is largely alleviated. After each compression, the newly reached local optima tend to yield further gains, enabling more consistent progress and higher final performance. These results confirm that the information loss introduced by compression helps escape local optima, restructuring the optimization landscape and allowing more continued and effective updates.

**Analysis of Prompt Length** To further understand how information loss impacts prompt performance, Figure 5d tracks the evolution of performance and prompt length during optimization. In two GRACE curves, local performance peaks often coincide with local maxima in prompt length, suggesting that adding reined information can enhance task-solving abilities. However, information quantity and performance are not necessarily positively correlated. At several compression points, performance remains stable or even improves despite a shorter prompt, indicating that concise, distilled instructions may be more effective than redundant, detailed ones. Moreover, in two curves without compression, performance stagnates as prompt length increases. As shown in Section 3.5, this growth in prompt length often correlates with the accumulation of increasingly case-specific details. This suggests that such information, while seemingly informative, typically lacks general utility and yields no gains. By strategically losing information in update signals and prompts, GRACE stably generates improved updates and escapes local optima, achieving more effective and sustained prompt optimization.

## 3.4 Ablation Study: Hyperparameter Selection

Our GRACE method aims to achieve significant performance improvements with much lower overhead. It relies on three key hyperparameters: (1) the number of correct and incorrect samples used per update, denoted as $S_t{}'$ and $F_t{}'$; (2) the maximum number of optimization steps $T$; (3) the number of consecutive rejections $K$ required to trigger compression. To justify our choices and guide hyperparameter settings on new tasks, we present the performance of varying $T$ and $K$ in Figure 6, and report ablations on $S_t'$ and $F_t'$ in Figure 5a and Figure 7.

**Selection of the Optimization Steps** We set $T = 80$, which provides optimal performance for most tasks, achieving a balance between performance and cost. Increasing $T$ generally improves performance, but with diminishing returns, while computational cost and resource requirements grow linearly. As shown in Figure 6, $T = 80$ achieves the best results on three tasks and near-optimal performance on the remaining two. Further increasing $T$ yields minimal improvements but incurs a linear increase in cost. Thus, to balance performance and efficiency, we use $T = 80$ as the default. Additionally, we observe that when no improvement is seen over 20 consecutive steps, more iterations rarely help. Therefore, on new tasks, we recommend setting $T = 80$, possibly combined with an early stopping criterion of 20 stagnant steps.

**Selection of the Compression Trigger** We set $K = 5$, as it provides optimal performance across most tasks. As shown in Figure 6, both smaller and larger values of $K$ lead to decreased performance.

| State | Prompt | Score |
|---|---|---|
| Step 0 Initial | Read carefully the following premise and hypothesis, and determine the relationship between them. Choose from 'contradiction', 'neutral', or 'entailment'. | 89.3 |
| Step 1 Parent 0 | Read ... Determine their relationship by analyzing explicit statements, including the speaker's perspective and pronoun references. Choose from ... | 91.1 |
| Step 2 Parent 1 | Read ... Determine their relationship by analyzing whether the premise directly supports (entailment), contradicts, or neither (neutral). Pay attention to: Whether statements are presented as facts, hypotheticals, or opinions; Whether questions or possibilities in the premise justify the hypothesis. | 92.9 |
| **Step 3** Parent 2 | Read ... Analyze both explicit statements and implicit implications, including those from conditional clauses, presuppositions, or references. Determine if the premise necessarily supports (entailment), directly contradicts (contradiction), or neither (neutral). | **94.6** |
| Step 4 Parent 3 | Read ... Analyze both explicit statements and implicit implications, including those from conditional clauses (noting their pragmatic implications), presuppositions, and references (resolving coreference and speaker identity). Distinguish between factual assertions and subjective opinions. Determine if ... | 92.9 |
| Step 5 Parent 3 | Read ... Analyze both ... Distinguish between assertions of belief/opinion and objective facts. For conditional statements, evaluate whether the premise provides evidence beyond hypothetical scenarios. When resolving references or coreferences, rely solely on explicit information. Determine if ... | 91.1 |
| Step 9 Parent 3 | Read ... Analyze both explicit statements and implicit implications to determine if the premise entails, contradicts, or is neutral toward the hypothesis. | 94.6 |
| **Step 13** Parent 12 | Read ... Analyze both explicit statements and implicit implications, including beliefs, hypothetical scenarios, and conditional statements. Determine if ... by evaluating factual support, direct opposition, or lack of relevant information. | **96.4** |

Table 2: Prompt optimization process on CB task. In **State**, green, red and blue denote whether the current prompt is updated, rejected, or compressed from the parent prompt, respectively. In **Prompt**, green, red and blue mark modification over parent prompts ,which is beneficial (leading to update), harmful (leading to rejection) or compressed, respectively. Step 3 and Step 13 is local optimum.

A small $K$ may trigger premature compression before sufficient optimization has been explored, destabilizing the optimization process. On the other hand, a large $K$ may provide more optimization opportunities but could also waste resources, reducing the number of effective optimization in a limited number of iterations, leading to lower final performance. Therefore, we suggest set $K = 5$ to achieve a balance between exploration and exploitation in the prompt optimization space.

### 3.5 Qualitative Analysis

To further vividly show how GRACE turns information loss into performance gains, we conduct a qualitative analysis of the optimized trace on CB task. Table 2 presents instances of accepted updates, rejections, and compression steps, along with corresponding prompt text changes and validation scores. Modifications relative to the parent prompt are highlighted, distinguishing helpful, harmful, and compressed content using different colors. In the initial phase (**Step 0 to Step 3**), GRACE enhances the prompt by refining existing instructions and incorporating new task-relevant information, leading to steady performance improvements. However, in the later steps (**Step 4 and Step 5**), the performance declines despite the more enriched prompt. This illustrates a typical case of getting trapped in a local optimum: although the prompt becomes more elaborate, the added information increasingly consists of overly specific, case-bound logic, which offers little utility for unseen examples and can even degrade performance. When such a stagnation is detected, GRACE adaptively compresses the current prompt while preserving its essential guidance. In **Step 9**, the adaptive compression introduces information loss by removing redundant specifics, yet retains the core instructional content, resulting in no performance degradation. Later, in **Step 13**, a new local optimum built on the compressed version, the prompt incorporates more generalizable and valuable guidance, surpassing the earlier peak at Step 3 and yielding further performance gains. These observations highlight the value of compression in helping escape local optima. By resetting the optimization trajectory, compression facilitates global exploration and enables more effective and sustained local exploitation. To facilitate a clearer comparison with other methods, we include the

prompt optimization process for OPRO (as a representative search-based method) and APO (as a representative reflection-based method) in Appendix C.

## 3.6 Cost Analysis

Beyond task performance, the computational cost of prompt optimization is also a key concern [44]. Table 3 compares the base and optimizer LLM costs of GRACE against baseline methods on TREC dataset. Detailed token and API usage for both the input and output are provided in Table 11 (Appendix D). For base LLM cost, all baseline methods except APO are expensive, as they generate and evaluate numerous candidates at each step. Although APO employs a UCB algorithm to reduce base LLM evaluations [2], it still relies on optimizer LLM to generate a large number of candidates, resulting in high cost on the optimizer side. Search-based methods such as EvoPrompt and OPRO lack clear optimization guidance and often produce short prompts with low token costs, leading to relatively low optimizer LLM cost, but at the expense of limited performance gains. In contrast, GRACE conducts more targeted optimization, resulting in low cost on both the base and optimizer LLMs, achieving superior performance with significantly lower overall costs.

|  | Base | Opt | Sum | Score |
|---|---|---|---|---|
| EvoPrompt | 6.5 | **0.5** | 7.0 | 85.4 |
| OPRO | 7.3 | 1.3 | 8.6 | 86.4 |
| APO | **2.0** | 1.6 | 3.6 | 90.6 |
| PromptAgent | 6.6 | 3.9 | 10.5 | 90.2 |
| GRACE | **2.0** | 0.8 | **2.8** | **94.2** |

Table 3: Cost($) comparison of base and optimizer LLMs on the TREC task.

## 4 Related Works

**Automatic Prompt Engineering** Automatic prompt engineering has emerged as a lightweight alternative to full fine-tuning for adapting LLMs to downstream tasks [13, 26]. One line of work uses reinforcement learning to train auxiliary editing agents that iteratively refine discrete prompts (at the token, phrase, or sentence level) based on reward signals derived from task performance [7, 50, 34, 9, 18, 49]. Other discrete methods apply gradient-based search to directly optimize token sequences [30, 42]. Another direction focuses on tuning soft prompts, which are learnable continuous embeddings prepended to input sequences, offering a parameter-efficient adaptation strategy [19, 14, 39, 27]. However, these methods typically require access to the model's internal states or gradients, making them inapplicable to closed-source API-based LLMs.

**LLM-based Prompt Optimization** Recent approaches to automate prompt optimization for closed-source models employ LLMs as optimizers to iterative expand and select prompt candidates [51], which can be broadly categorized into two types based on their expansion strategies. Search-based methods update prompts using heuristics, including phrase deletion or swapping [24], back-translation [45], and evolutionary algorithms [11, 12]. Reflection-based methods revise prompts by analyzing model errors and generating natural language feedback from failed examples to guide updates [25, 47, 48, 44]. To improve global optimization, various search algorithms have been integrated, including Monte Carlo sampling [51], Gibbs sampling [46], beam search [25], and Monte Carlo Tree Search [37]. GRACE distinguishes itself by introducing gated refinement and adaptive compression strategies to more stably improve prompts and escape local optima, leading to more efficient and effective optimization. While [43] also utilize success samples in prompt optimization, their primary purpose is to mitigate forgetting. In contrast, GRACE goes further by meticulously analyzing their regularization role and leveraging them to flexibly control update signals from failed examples.

## 5 Conclusion

In this paper, we introduce GRACE, a novel prompt optimization framework that leverages information loss to achieve efficient prompt optimization and overcome getting trapped in local optima. GRACE integrates two synergistic strategies, gated refinement and adaptive compression, which work in coordination to support fine-grained local updates and periodic global restructuring. Extensive experiments on 11 tasks across three practical domains demonstrate GRACE's substantial gains in both performance and efficiency. It significantly outperforms prior state-of-the-art methods, while requiring notably less prompt generation and incurring lower computational overhead. We believe that GRACE offers a promising direction for advancing the future of prompt engineering.

# 6 Acknowledgements

The work was supported in part by the National Natural Science Foundation of China (NSFC) under Grant 62461146205, 62441230, 62502522 and 62072458, and in part by the Outstanding Innovative Talents Cultivation Funded Programs 2024 of Renmin University of China.

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

# A   Implementation Details

## A.1   Tasks and Datasets

To comprehensively evaluate the effectiveness of GRACE, we curate 11 tasks spanning three distinct categories: BIG-Bench Hard (BBH), domain-specific expert tasks, and general NLP tasks. BBH tasks [35] represent a challenging subset of the broader BIG-Bench benchmark [32], designed to push the capabilities of modern LLMs. Considering the continuous improvement in LLM performance, we specifically select 5 BBH tasks where our base model, DeepSeek-V3-0324, still struggles when using the original human-provided task instructions. Moreover, we select three domain-specific tasks from the biomedical domain: information extraction (NCBI [8]), sentence similarity (Biosses [31]), and question answering (Med QA [15]). Beyond challenging and domain tasks, to further demonstrate that GRACE can enhance performance on traditional NLP tasks, we select three well-known NLU tasks, i.e., TREC [36], Subj [23], and CB [4]. Accuracy is used as the evaluation function for all tasks, except for the NCBI task, where the F1 score is employed.

| Task | Train | Valid | Test |
|------|-------|-------|------|
| **BigBench** | | | |
| Geometric Shapes | 150 | 70 | 200 |
| Salient Translation Error | 110 | 60 | 140 |
| Snarks | 82 | 45 | 95 |
| Movie Recommendation | 110 | 60 | 140 |
| Epistemic | 400 | 160 | 400 |
| **Domain Knowledge** | | | |
| NCBI | 1000 | 500 | 940 |
| Biosses | 60 | 30 | 40 |
| Med QA | 200 | 100 | 400 |
| **General NLP** | | | |
| Subj | 400 | 150 | 1000 |
| TREC | 400 | 150 | 500 |
| CB | 125 | 65 | 56 |

Table 4: Data split.

**Data Split** For datasets that provide predefined test sets, we utilize these directly for our final test evaluation, capping the size at a maximum of 1000 samples. If a default test set is not available, we randomly shuffle the entire dataset and allocate approximately half of the samples for testing. The remaining data constitutes the training set, from which we randomly select a small, non-overlapping subset to serve as the validation set used during the prompt optimization process. The specific data splits for each task are detailed in Table 4.

**Prompt Initialization** All optimization methods start with the same initial prompt, a manual task instruction, Task (ZS), with the exception of EvoPrompt. For tasks that natively include task instructions (e.g., BBH), we use these directly. For other tasks, we construct or collect suitable initial instructions from established resources like PromptSource [3] or Natural Instructions [38]. EvoPrompt requires 14 additional varied instructions. If the original EvoPrompt code dose not provide these for a specific task, we generate them by paraphrasing the initial human instruction. The specific initial prompt used for each task is detailed in Tables 12 to 16.

## A.2   Method Implementation Details

GRACE and all baseline methods use an optimizer LLM to generate the candidate prompts. Based on recommendations from official documentation and technical reports, the temperature for the optimizer LLM is set to 0.6, while the temperature for the base LLM (the model executing the downstream task) is set to 0. To fairly compare the performance of different methods, we ensure that the number of prompt searches for each baseline method is approximately 300 and generally sufficient for convergence. This budget follows the original parameter settings for each method and the comparative experimental setups described in [21] and [43]. The implementations for all baseline methods are based on their respective official code releases, with modifications made to align the evaluation budgets where necessary. In addition, to bolster the reliability of results, any experiments that yield anomalous results or significant deviations from expected performance are re-conducted. We illustrate the details for various baselines and our GRACE methods in our experiments, with specific parameter configurations provided in Table 5.

**EvoPrompt** [12]. EvoPrompt introduces evolutionary algorithms into the prompt optimization process. It initializes with a population of 15 prompts. In each of its 10 steps, it applies evolutionary operators (e.g., mutation, crossover) to the current population to generate 30 candidate prompts. In

**Algorithm 1** GRACE Framework Overview

---

**Require:** Initial prompt $\mathcal{P}_0$, Dataset $D$, optimization function $p_{\mathcal{O}}$, evaluation function $f_{\mathcal{B}}$
**Ensure:** Optimal Prompt $\mathcal{P}^*$
1: $reject\_counter \leftarrow 0$
2: **for** $t = 0$ to $T$ **do**
3:     # Gated Refinement
4:     Partition $D_{train}$ into $S_t, F_t$ based on $f_{\mathcal{B}}(\mathcal{P}_t, D_{train})$
5:     Sample update batch $B_t = S_t' \cup F_t'$, ( $S_t' \subseteq S_t$ and $F_t' \subseteq F_t$)
6:     Generate candidate $\mathcal{P}_t^c \sim p_{\mathcal{O}}(\mathcal{P} \mid \mathcal{P}_t, B_t, m_1)$          ▷ Feedback Regulation Gate
7:     Update $\mathcal{P}_{t+1} = argmax_{\mathcal{P} \in \{\mathcal{P}_t, \mathcal{P}_t^c\}} f_{\mathcal{B}}(\mathcal{P}, D_{val})$      ▷ Update Rejection Gate
8:     # Adaptive Compression
9:     **if** $\mathcal{P}_{t+1} = \mathcal{P}_t$ **then**
10:         $reject\_counter \leftarrow reject\_counter + 1$
11:     **else**
12:         $reject\_counter \leftarrow 0$
13:     **end if**
14:     **if** $reject\_counter = K$ **then**
15:         $\mathcal{P}_{t+1} \sim p_{\mathcal{O}}(\mathcal{P} \mid \mathcal{P}_t, m_2)$
16:         $reject\_counter \leftarrow 0$
17:     **end if**
18: **end for**
19: **return** $\mathcal{P}^*$ with best $f_{\mathcal{B}}(\mathcal{P}, D_{val})$

---

our experiments, we use the genetic algorithm. Based on validation set performance, it selects the top 15 prompts to form the population for the next generation.

**OPRO** [47]. OPRO incorporates the optimization trajectory (historical prompts and scores) into its process. In each of its 20 rounds, it uses a meta-prompt containing information from the top 20 prompts evaluated so far (based on validation performance) to generate 15 new candidate prompts.

**APO** [25]. APO models prompt optimization as a beam search process. With a beam size of 5, each step involves reflecting on the errors associated with each prompt currently in the beam. This reflection guides the generation of 5 improved versions and 6 paraphrased versions for each prompt. All generated candidates are evaluated on the validation set, and the top 5 overall performers are retained in the beam for the next round. This process runs for 6 rounds.

**PromptAgent** [37]. PromptAgent formulates prompt optimization as a planning problem solved via a Monte Carlo Tree Search (MCTS) framework. We modify its standard configuration to an expansion width of 4, a depth limit of 10, and 12 MCTS iterations per prompt generation step, leading to a comparable evaluation budget.

**GRACE**. In contrast to prior work that explores numerous prompts per step primarily to mitigate instability, our method, GRACE, generates only a single prompt in each step. This new prompt is either a candidate update from gated refinement or a compressed version. It runs for a maximum of 80 iterations, which is sufficient for convergence across all datasets, requiring fewer total prompt evaluations than the baselines. The detailed algorithmic procedure is presented in Algorithm 1.

# B   Additional Experiment Results

## B.1   Detailed and Additional BBH Results

Table 1 presents a performance comparison of various methods across three task categories, where only the average performance achieved using DeepSeek-V3-0324 as the base LLM is shown for the Big-Bench Hard (BBH) tasks. In Appendix Table 6, we present detailed results for each individual BBH task, again using DeepSeek-V3-0324 as the base LLM. The results show that GRACE consistently and significantly outperforms static prompts and other prompt optimization methods across all evaluated tasks. Furthermore, to demonstrate GRACE's generalizability, we conduct additional experiments

| Methods | Official Search Strategy | | | | Prompt Updating | Our Experiments Settings | | | | |
|---|---|---|---|---|---|---|---|---|---|---|
| | Initial size | Expansion size per step | Candidate size per step | Total Steps | Method Type | Initial size | Expansion size per step | Candidate size per step | Total Steps | Total Search |
| **EvoPrompt** | 10 | 10 | 10 | 10 | Evolution Algorithm | 15 | 30 | 15 | 10 | 300 |
| **OPRO** | 1 | 8 | – | 200 | Implicit Reflection | 1 | 15 | – | 20 | 300 |
| **APO** | 1 | $|P_{t-1}| \times 12$ | 4 | 6 | Explicit Reflection | 1 | $|P_{t-1}| \times 11$ | 5 | 6 | 286 |
| **PromptAgent** | 1 | – | – | 3 | Explicit Reflection | 1 | – | – | 12 | – |
| **GRACE** | – | – | – | – | Reflection & Compression | 1 | 1 | 1 | 80 | 80 |

Table 5: Parameter configurations of existing prompt optimization methods in our comparisons. "–" means the setting is not applicable to the method (in the case of PromptAgent, the search size per step is associated with the real-time process of MCTS). $|P_{t-1}|$ denotes the set of the prompts to be updated at each step t.

| Base LLM | Method | Geometry | Translation | Snarks | Movie | Epistemic | Avg. |
|---|---|---|---|---|---|---|---|
| DeepSeek-V3-0324 | Task (ZS) | 65.50 | 65.71 | 85.26 | 78.57 | 92.20 | 77.45 |
| | Task (FS) | 62.00 | 70.71 | 83.16 | 79.29 | 68.50 | 72.73 |
| | CoT (ZS) | 80.00 | 66.43 | 86.32 | 70.71 | 85.25 | 77.74 |
| | CoT (FS) | 56.00 | 75.00 | 88.42 | 91.43 | 87.25 | 79.62 |
| | EvoPrompt | 76.50 | 72.86 | 87.37 | 79.29 | 89.75 | 81.15 |
| | OPRO | 82.50 | 71.43 | 90.53 | 93.57 | 89.50 | 85.51 |
| | APO | 84.00 | 79.29 | 90.53 | **97.14** | 89.75 | 88.14 |
| | PromptAgent | 88.50 | 77.86 | 92.63 | 92.63 | 95.50 | 89.42 |
| | GRACE | **97.00** | **84.29** | **94.74** | **97.14** | **97.50** | **94.13** |
| GPT-4.1 | Task (ZS) | 43.00 | 72.86 | 93.68 | 75.00 | 87.00 | 74.31 |
| | Task (FS) | 70.00 | 73.57 | 82.11 | 83.57 | 81.25 | 78.10 |
| | CoT (ZS) | 40.00 | 70.71 | 94.74 | 70.00 | 87.50 | 72.59 |
| | CoT (FS) | 75.00 | 75.00 | 91.58 | 86.43 | 89.50 | 83.50 |
| | EvoPrompt | 70.00 | 76.43 | 93.68 | 82.14 | 90.50 | 82.55 |
| | OPRO | 65.00 | 76.43 | 94.74 | 90.00 | 88.00 | 82.83 |
| | APO | 88.00 | 78.57 | **95.79** | 92.86 | 90.50 | 89.14 |
| | PromptAgent | 85.00 | 80.00 | **95.79** | 94.62 | 92.00 | 89.48 |
| | GRACE | **94.50** | **85.00** | **95.79** | **97.14** | **96.50** | **93.79** |

Table 6: Detailed performances of different methods on five BBH tasks: Geometric Shapes (Geometry), Salient Translation Error Detection (Translation), Snarks, Movie Recommendation (Movie), Epistemic. Results are shown separately for DeepSeek-V3-0324 and GPT-4.1 as base LLMs. Bold text indicates the best performance achieved.

using GPT-4.1 [2] as the base LLM. On this more advanced model, GRACE again achieves significant performance improvements compared to baseline methods. This demonstrates that GRACE's key strategies work effectively with different base LLMs, highlighting the framework's broad applicability.

## B.2 Transferability of Optimized Prompts Across Base LLMs

Since base LLMs differ in architecture, pretraining data, and instruction tuning, we evaluate whether prompts optimized for one model generalize to others. Specifically, we optimize prompts using DeepSeek-V3-0324 as the target base LLM on five BBH tasks, then evaluate them on Llama-3.3-70B-Instruct [10] and GPT-4.1 [1] without further tuning. As shown in Table 7, GRACE-optimized prompts usually outperform both the initial prompts and those from PromptAgent, indicating transferability. However, the performance gains are highest on the model used for optimization (DeepSeek-V3) and generally smaller when transferred to other models. In a few instances (e.g., Epistemic on GPT-4.1), the optimized prompts even underperform compared to the initial ones. These results suggest that while GRACE prompts exhibit partial transferability, the optimized prompt is most effective when applied to the target base LLM.

---

[2]https://openai.com/index/gpt-4-1

| Task | DeepSeek-V3 | | | LLaMA3.3-70B | | | GPT-4.1 | | |
|------|------|------|-------|------|------|-------|------|------|-------|
| | ZS | PA | GRACE | ZS | PA | GRACE | ZS | PA | GRACE |
| Geometry | 62.50 | 88.50 | **97.00** | 72.50 | 70.00 | **75.50** | 43.00 | 50.00 | **52.00** |
| Translation | 65.71 | 77.86 | **84.29** | 70.00 | 70.71 | **72.86** | 72.86 | 77.14 | **80.71** |
| Snarks | 85.26 | 92.63 | **94.74** | **92.63** | **92.63** | **92.63** | **93.68** | 89.47 | **93.68** |
| Movie | 78.57 | 92.63 | **97.14** | 69.29 | 80.00 | **92.86** | 75.00 | 82.86 | **92.14** |
| Epistemic | 92.20 | 95.50 | **97.50** | 85.25 | 87.00 | **92.00** | **87.00** | 84.25 | 82.50 |
| Average | 77.45 | 89.42 | **94.13** | 77.93 | 80.07 | **85.17** | 74.31 | 76.74 | **80.21** |

Table 7: Transfer performance of prompts optimized with DeepSeek-V3 as the base LLM across other models. ZS denotes the task's zero-shot initial prompt; PA denotes prompts optimized by PromptAgent. Bold indicates the best result for each task–model pair.

| Method | Reddit | Amazon | Avg. |
|--------|--------|--------|------|
| Task(ZS) | 12.21 | 16.78 | 14.50 |
| Task(FS) | 12.64 | 18.52 | 15.58 |
| EvoPrompt | 13.24 | 19.24 | 16.24 |
| OPRO | 13.13 | 20.35 | 16.74 |
| APO | 14.33 | 21.45 | 17.89 |
| PromptAgent | 16.29 | 21.29 | 18.79 |
| GRACE | **17.60** | **23.76** | **20.68** |

Table 8: Performance on the summarization task, measured by ROUGE-L. ZS/FS denote Zero-Shot and Few-Shot settings. Task (ZS) is the initial prompt for prompt optimization methods. Bold values indicate the best in each column.

## B.3 Additional Experiments on Summarization Tasks

In Table 1, we demonstrate the effectiveness of our GRACE method on complex reasoning, domain-specific, and natural language understanding tasks, which are the most common and standard benchmark tasks for automatic prompt optimization methods. To assess broader applicability of our method, we conduct experiments on two summarization tasks: the Reddit TIFU dataset [16] and the Amazon Fine Food Reviews dataset [33]. Task instructions come from Super-NaturalInstructions [38], and performance is measured by Rouge-L. To adapt prompt optimization methods to summarization tasks, we map outcomes to a binary signal by labeling the top 20% Rouge-L instances as "correct" and the bottom 20% as "incorrect." As shown in Table 8, GRACE achieves the best performance on both datasets, indicating strong effectiveness on summarization and generalization beyond classification and reasoning tasks.

## B.4 Detailed Ablation on Feedback Regulation Gate

Figure 7 presents a detailed ablation analysis of the feedback regulation gate in gated refinement strategy. Specifically, we analyze the optimization performance when using an update batch size of 6, varying the number of success samples from 0 to 5 (and failure samples correspondingly from 6 to 1). The left plot depicts scenarios with fewer than three success samples. In these cases, we observe similar trends: rapid initial convergence to a local optimum, followed by performance fluctuations. Increasing the number of success samples reduces the fluctuation magnitude and increases the likelihood of escaping the local optimum via the adaptive compression. Observing the right plot, which shows cases with three or more success samples, the optimization paths appear more stable. Performance tends to increase slowly but steadily with the number of explored prompts. However, an excessive proportion of success samples can be counterproductive. By overly slowing optimization, it leads to minimal updates that may impede progress in the discrete prompt space, ultimately limiting the achievable performance. These observations align with our main ablation results, confirming that success samples act as a regularizer in the feedback regulation gate. Their proportion in the update batch can effectively control the trade-off between optimization speed and stability.

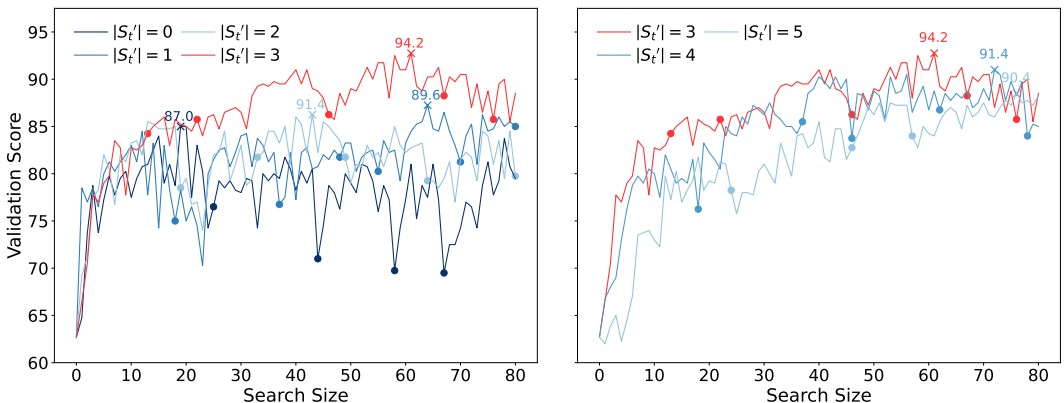

Figure 7: A more detailed version of Figure 5a, presenting convergence curves of setting $|S_t'|$ from 0 to 5.

| State | Prompt | Score |
|---|---|---|
| Step 0 Initial | Read carefully the following premise and hypothesis, and determine the relationship between them. Choose from 'contradiction', 'neutral', or 'entailment'. | 89.3 |
| Step 1 Parent 0 | Examine the premise for explicit statements or logical conclusions that directly support the hypothesis. Determine the relationship as contradiction, neutral, or entailment. | 89.3 |
| Step 4 Parent 0 | First, determine if the hypothesis is a logically necessary conclusion of the premise (entailment). If the premise directly negates or explicitly opposes the hypothesis, choose contradiction. If neither condition is definitively met, select neutral. | 91.1 |
| Step 7 Parent 4 | First, check if the premise definitively confirms the hypothesis (entailment). If the premise directly negates or makes the hypothesis logically impossible, select contradiction. If neither condition applies, choose neutral. | 89.3 |
| Step 16 Parent 4 | First, confirm if the hypothesis is explicitly stated or irrefutably entailed by the premise—select entailment. If the premise directly contradicts or logically invalidates the hypothesis, choose contradiction. If neither condition is definitively met, select neutral. | 91.1 |
| Step 20 Parent 4 | First, ascertain whether the hypothesis is explicitly stated or logically inescapable from the premise (entailment). If the premise explicitly negates the hypothesis or renders it logically invalid, select contradiction. When neither definitive confirmation nor definitive refutation exists without external inference, choose neutral. | 89.3 |

Table 9: Prompt optimization process of OPRO method on CB task. At each step, OPRO generates 15 new candidates based on top-performing prompts. We show the top prompt at selected steps. In **State**, green indicates that the current prompt is better than the parent one. Step 4 and Step 16 is local optimum.

## C  Prompt Optimization Process of Baseline Methods

In Section 3.5, we present a qualitative analysis of GRACE's optimized trace on the CB task, showing how it turns information loss into performance gains. For a clearer comparison with baselines, we additionally report the prompt optimization trajectories of representative methods. We select OPRO as a search-based baseline; its optimization process is shown in Table 9. As observed, OPRO's candidates at each step largely preserve the original semantics, yielding limited diversity and slow improvements. For reflection-based methods, we use APO; its trajectory appears in Table 10. In the first step, APO mainly adds general guidance with a few specific details, producing a gain. Subsequent revisions, however, focus on narrowly tailored logic or memorized phrasings tied to particular examples, which fail to generalize and can even degrade performance.

| State | Prompt | Score |
|---|---|---|
| Step 0
Initial | Read carefully the following premise and hypothesis, and determine the relationship between them. Choose from 'contradiction', 'neutral', or 'entailment'. | 89.3 |
| **Step 1**
Parent 0 | Analyze the given premise and hypothesis step by step:\n1. Identify explicit statements, negations (e.g., 'don't', 'never'), uncertain language (e.g., 'might', 'I think'), hypotheticals, or questions in the premise.\n2. Determine if the hypothesis is directly supported (entailment), directly opposed (contradiction), or neither (neutral) based on these elements.\n - For contradictions: Check if the premise explicitly denies the hypothesis or expresses doubt about it (e.g., 'I don't think X' contradicts 'X will happen').\n - For entailment: Verify the premise provides clear, unambiguous support for the hypothesis.\n - For neutral: Apply if the premise is uncertain, asks a question without an answer, or lacks information to confirm/deny the hypothesis.\n3. Avoid assuming unstated intentions (e.g., a question like 'Should they spend more?' does not entail 'they should spend more' unless explicitly agreed). \n\n Relationship options: contradiction, neutral, entailment. | **94.6** |
| Step 2
Parent 1 | Assess the connection between a premise and hypothesis using structured evaluation: \n\n1. Examine language and reasoning indicators: \n - Direct assertions: Recognize overt claims, denials (e.g., "cannot"), or confirmations. \n - Indirect links: Uncover implied logic (e.g., modus ponens: given "If A, then B" and A, infer B), assumptions, or persuasive techniques (e.g., rhetorical questions suggesting answers). \n\n2. Determine the connection type: \n - Contradiction: ...(e.g., "X is untrue" or "If X, then ¬Y" with X verified). \n - Entailment: ...(including via conditionals or rhetorical cues). \n - Neutral: ... \n\n3. Protocols: \n - Conditionals: Interpret "If X, then Y" as entailment if X is validated in the premise. \n - Rhetorical devices: Treat questions like "Wasn't X agreed?" as assertions of X's truth. \n - Limit inferences: Base conclusions only on stated or logically derived information. \n\nIllustrations:\n- Premise: "Should Lumina inquire, we'll acknowledge Verdant is present." Hypothesis: "Verdant is present." → Entailment (conditional agreement). \n- Premise: "Hadn't I stated Azure is the meeting site?" Hypothesis: "Azure is where they convened." → Entailment (rhetorical confirmation). \n\nCategories: ... | 92.9 |
| Step 6
Parent 1 | Determine ... by evaluating factual consistency, negation implications, and contextual alignment. \n\n Classification Rules \n1. Contradiction: ... Explicit Factual Opposition: Clear factual conflicts (e.g., "The road is dry" vs. "The road is wet"). \n - Logical Incompatibility: Premise creates conditions that invalidate the hypothesis. \n - Focus on factual clashes, not subjective disagreements (e.g., rejecting a belief ≠ rejecting the hypothesis). \n\n2. Entailment: ... Explicit Confirmation: Premise directly states or logically guarantees the hypothesis. \n - Contextual Support: Premise offers clear real-world validation (e.g., "He confirmed the event occurred"). \n - Definitions alone ≠ support unless explicitly tied to the hypothesis. \n\n3. Neutral: ... Non-Actionable Statements: Beliefs, assumptions, or emotions about the hypothesis ≠ proof (e.g., "I suspect X" ≠ "X is true"). \n - Isolated Definitions: Explaining terms without applying them to the hypothesis**. \n - Non-Committal Queries: Questions like "Do you think X?" ≠ factual claims. \n\n Key Considerations: \n- Negation Implications: \n - Premise negating intent (e.g., "He didn't plan to go") ≠ contradiction of the action itself unless the action's occurrence is denied. \n- Definitions vs. Assertions: \n - Defining terms (e.g., "X refers to Y") ≠ entailment unless paired with a factual claim about the hypothesis. | 91.1 |

Table 10: Prompt optimization process of APO method on CB task. At each step, APO generates 55 new candidates from the previous round's prompts. We show the top prompt at selected steps. In **State**, green indicates improvement over the previous round, red indicates degradation. In **Prompt**, edits relative to the parent are highlighted in green (beneficial) and red (harmful). Step 1 reaches a local optimum.

| | Base LLM | | | Optimizer LLM | | | | |
|---|---|---|---|---|---|---|---|---|
| | API | Tokens$_I$ | Tokens$_O$ | API | Tokens$_I$ | Tokens$_O$ | Cost ($) | Score |
| EvoPrompt | 45K | 3.8M | 5.4M | 318 | 0.1M | 0.2M | 7.0 | 85.4 |
| OPRO | 45K | 6.9M | 5.7M | 300 | 0.4M | 0.5M | 8.6 | 86.4 |
| APO | 9.7K | 5.6M | 1.0M | 574 | 0.6M | 0.6M | 3.6 | 90.6 |
| PromptAgent | 33.6K | 24.3M | 2.7M | 448 | 3.0M | 1.1M | 10.5 | 90.2 |
| GRACE | 14.4K | 6.9M | 0.9M | 80 | 0.3M | 0.3M | 2.8 | 94.2 |

Table 11: Cost comparison on the TREC task (I: Input, O: Output).

# D    Cost Analysis

During the execution of each automatic prompt optimization method, we record the number of API calls and the input/output token counts for both the base LLM and the optimizer LLM. Based on the respective API pricing models [3], we further calculate the estimated cost per run. Note that our cost calculation for the optimizer LLM (DeepSeek-R1) does not consider the reasoning process. Table 11 presents a detailed, fine-grained comparison of this resource consumption across the different methods. While most API calls and tokens are used for evaluating prompts with the base LLM, the optimizer LLM calls also significantly impact total cost because they process more tokens and have a higher price per call. Therefore, by performing more efficient optimization, GRACE achieves better performance with lower resource costs.

# E    Limitations

## E.1    Fair Comparison with Baseline Methods

To ensure a fair comparison with baseline methods, we align the maximum number of generated prompts for all baselines to approximately 300, consistent with comparative experiments in prior works [21, 43]. Although we observe convergence for existing methods across all datasets within this limit, we cannot guarantee that every method reaches its absolute peak performance, as some search-based approaches like OPRO are originally designed for a much larger search budget (e.g., 1600 prompts). Given that optimization cost and efficiency are crucial factors alongside final performance, and noting that GRACE utilizes a maximum of only 80 prompt evaluations, we believe this search limit of approximately 300 evaluations for baselines is reasonable for a fair comparative assessment.

## E.2    Base Model Selection

In our experiments, the DeepSeek-V3-0324 model is primarily selected as the base model. Although DeepSeek-V3-0324 is a current state-of-the-art model, a potential concern is that the performance ceiling of LLMs is continually advancing, and different LLMs may exhibit varying characteristics, potentially limiting the generalizability of our method and the findings. To address this concern, we conduct additional experiments using GPT-4.1 as the base model (details in Appendix B.1). GRACE again achieves consistent and significant performance improvements, demonstrating the general applicability of our method. Furthermore, for base models in the context of automatic prompt optimization, a key capability influencing results is instruction-following, which is consistently improving in newer LLMs. Therefore, we are confident that GRACE will remain competitive and effective as newer, more sophisticated base models become available.

## E.3    Limits on Tasks Requiring Specialized Knowledge

We have demonstrated that GRACE achieves strong performance on complex reasoning, natural language understanding, and domain-specific tasks, significantly surpassing existing methods in both efficiency and effectiveness. However, upon further analysis, we find that the performance gains on tasks requiring specialized domain knowledge are relatively limited. As shown in Table 1, the lift on MedQA is modest. Our failure case analysis reveals that certain samples in the MedQA

---

[3]https://api-docs.deepseek.com/quick_start/pricing

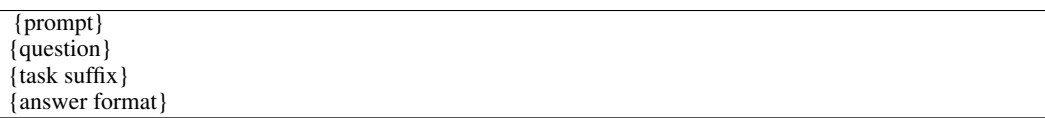

| |
| --- |
| {prompt} |
| {question} |
| {task suffix} |
| {answer format} |

Table 12: Input prompt to base LLM.

dataset require highly specialized medical knowledge that neither the optimizer nor the base LLMs possess. This indicates that our method struggles to effectively address such cases. It is important to emphasize that this limitation is not unique to our method but reflects a gap in current automatic prompt optimization approaches, which all rely solely on the capabilities of the optimizer and base LLMs. To improve generalizability and practical utility on such tasks, we plan to augment GRACE with external knowledge access (e.g., retrieval-augmented generation). Enabling the optimizer to consult vetted domain repositories may supply the missing knowledge needed to construct more effective prompts for specialized settings.

## F Broader Impacts

Large Language Models (LLMs) are increasingly utilized across diverse industries, and effective prompting is crucial to fully leverage their capabilities. However, prompt engineering for closed-source models remains a complex and labor-intensive task, typically relying on human experts who must possess a deep understanding of both LLM behavior and task intricacies. Our method, GRACE, offers an effective and efficient approach to automatically generate effective gap-bridging prompts. This can significantly reduce reliance on human expertise, lower manual costs, and enable the efficient automation of a wider range of processes.

Beyond its positive impacts, GRACE could potentially have negative consequences. Because GRACE can discover effective prompts for practical tasks, there is a risk it could be exploited for malicious or unintended purposes, such as generating prompts for model jailbreaking. However, this particular risk is not unique to GRACE but is rather tied to the capabilities of the optimizer LLMs used in the process. Specifically, our GRACE method entails an optimizer LLM to guide the optimization. Therefore, to generate a prompt intended for unsafe applications, the optimizer LLM itself would first need to be capable of producing or engaging with unsafe content. This shifts the primary safety concern to the inherent safeguards and alignment of the optimizer LLM, rather than GRACE creating an entirely new vector for misuses.

## G Prompt Format

### G.1 Input Prompt

For all methods, the prompt format used as input to the base LLM is unified as follows:

$$\text{Input} = \text{Prompt} + \text{Question} + \text{TaskSuffix} + \text{AnswerFormat}.$$

The "Prompt" is our optimization target; the "Question" is the main body of the task's question; the "Task Suffix" is optional, including the options (For example, yes/no, entailment/non-entailment, or A, B... in tasks with multiple choices); and the "Answer Format" is designed for capturing answer from the model's response. We show the input format in Table 12, with one example for TREC task in Table 13.

### G.2 Meta-Prompt

In GRACE, we use different meta-prompts to let the optimizer LLM complete different tasks. We show the prompt format of input to the base LLM, error and correct strings, the prompt to update, and the prompt to simplify in Tables 12 to 16.

| |
|---|
| Tag the text according to the primary topic of the question. Choose from (A) Abbreviation, (B) Entity, (C) Description and abstract concept, (D) Human being, (E) Location, (F) Numeric value |
| Text: Who are the nomadic hunting and gathering tribe of the Kalahari Desert in Africa? Assign a label for the preceding text |
| Options: (A) Abbreviation (B) Entity (C) Description and abstract concept (D) Human being (E) Location (F) Numeric value |
| Put your answer option within \boxed{}. |

Table 13: One example of input prompt for TREC task.

| |
|---|
| <{index}> The model's input is: {question} The model's response (solution) is: {response} The correct label is: {label} The model's final prediction is: {prediction}. |

Table 14: Prompt of error or correct string for failed or successful cases.

# H Sensitivity to Meta-Prompt Design

Given the known sensitivity of LLMs to prompts, the design of meta-prompts can affect the effectiveness of prompt optimization. To evaluate the robustness of our method to meta-prompts, we paraphrase each key section of the meta-prompts used for optimization in Table 15 and compression in Table 16. For each section, we create 5 paraphrased variants and evaluate performance on 5 tasks from the BBH benchmark, reporting the mean and 95% confidence interval per task.

The meta-prompt for optimization consists of the following four components:

- **Main Task (MT)**: Your task is to optimize the current prompt for a language model performing a specific task. The goal is to correct previously failed predictions while preserving the model's correct behavior on already successful examples.
- **Preserve Correctness (PC)**: Ensure the model, instructed by the optimized prompt, continues to predict correct answers for all successful examples. In addition to prediction correctness, maintain the model's original correct solutions and response for these cases as much as possible.
- **Refine to Fix Errors (FE)**: For failed examples, attempt to correct them by refining the prompt's instructions — for example, by adding clearer or more complete guidance. Any new content should integrate naturally with the current prompt and form a coherent task instruction. Avoid special-case logic, examples, or instructions targeted at individual cases.
- **Additional Guidelines (AG)**: - Prompt modifications should always aim to preserve model's correct behavior on successful examples. ...

Performance with paraphrased versions of each component is shown in Table 17

The meta-prompt for compression contains two main components:

- **Main Task (MT)**: The prompt may have accumulated redundant, overly specific, or ineffective wording across previous iterations. Your goal is to ...
- **Additional Guidelines (AG)**: Eliminate instructions that are verbose, ambiguous, or unlikely to generalize ...

Performance with paraphrased versions of each component is shown in Table 18

Across all paraphrased variants, performance remains highly stable with minimal variance, demonstrating our method's strong robustness to variations in meta-prompts. In addition, this finding aligns with our observations in search-based prompt optimization methods, where capable LLMs are often insensitive to minor phrasing variations when the core intent is preserved.

Your task is to optimize the current prompt for a language model performing a specific task. The goal is to correct previously failed predictions while preserving the model's correct behavior on already successful examples.

The current prompt is:
"{current prompt}"

This prompt was evaluated on a batch of examples.
It successfully handled the following examples:
{correct string}
It failed on the following examples:
{error string}

Please analyze both the successful and failed examples.
Based on the example analysis, please optimize the current prompt under the following principles:
1. Preserve Correctness
Ensure the model, instructed by the optimized prompt, continues to predict correct answers for all successful examples. In addition to prediction correctness, maintain the model's original correct solutions and response for these cases as much as possible.
2. Refine to Fix Errors
For failed examples, attempt to correct them by refining the prompt's instructions — for example, by adding clearer or more complete guidance. Any new content should integrate naturally with the current prompt and form a coherent task instruction. Avoid special-case logic, examples, or instructions targeted at individual cases.

Additional guidelines:
- Prompt modifications should always aim to preserve model's correct behavior on successful examples.
- All changes should be minimal, necessary, and stable across iterations.
- The optimized prompt should be generalizable generalizable across different cases, rather than focusing on specific vocabulary or phrasing
- Only optimize the current prompt. Do not include input formats, verbalizers, or other fixed components.
- Provide the final optimized prompt within <START> and </START>.

Table 15: Meta-prompt 1: Updating the current prompt based on failed and successful cases.

Your task is to reconstruct a cleaner, more concise version of the current prompt for a language model.

The current prompt is: "{current prompt}"

The prompt may have accumulated redundant, overly specific, or ineffective wording across previous iterations. Your goal is to simplify and restructure it into a more effective and streamlined form — one that retains its core guidance while leaving room for future refinement.

Guidelines:
- Eliminate instructions that are verbose, ambiguous, or unlikely to generalize. Preserve the core intent and task framing, but express it as clearly and simply as possible.
- The new prompt should be self-contained, compact, and easy to iterate on in later optimization rounds.
- Provide the final optimized prompt within <START> and </START>.

Table 16: Meta-prompt 2: Compressing the current prompt

# I   Optimized Prompts from GRACE

We present the initial prompt and the optimized prompt from GRACE of different tasks on Tables 19 to 29.

| Method | Geometry | Translation | Snarks | Moive | Epistemic | Avg. |
|--------|----------|-------------|--------|-------|-----------|------|
| GRACE | 97.00 | 84.29 | 94.74 | 97.14 | 97.50 | 94.13 |
| MT | 97.00(0.71) | 84.29(1.51) | 94.32(1.60) | 97.14(0.51) | 97.45(0.37) | 94.04 |
| PC | 96.40(1.39) | 84.57(1.26) | 94.95(0.47) | 96.86(0.39) | 97.65(0.29) | 94.09 |
| FE | 97.00(1.00) | 83.57(1.34) | 94.95(1.15) | 96.86(0.39) | 97.35(0.60) | 93.95 |
| AG | 96.90(0.74) | 84.00(1.40) | 94.32(1.49) | 97.14(0.51) | 97.25(0.40) | 93.92 |

Table 17: Performance of paraphrased variants for different components of the meta-prompt used to update the prompt.

| Method | Geometry | Translation | Snarks | Moive | Epistemic | Avg. |
|--------|----------|-------------|--------|-------|-----------|------|
| GRACE | 97.00 | 84.29 | 94.74 | 97.14 | 97.50 | 94.13 |
| MT | 97.00(0.50) | 84.14(1.28) | 94.53(1.56) | 97.00(0.78) | 97.40(0.52) | 94.01 |
| AG | 97.29(1.64) | 84.00(0.96) | 94.74(1.49) | 97.28(0.32) | 97.55(0.33) | 94.17 |

Table 18: Performance of paraphrased variants for different components of the meta-prompt used to compress the prompt.

| |
|---|
| Extract the disease or condition from the sentence, if any is mentioned. |
| Extract all diseases and medical conditions from the text, including:
- Specific diagnoses, pathological states (e.g., abnormalities, disorders), and genetic disorders (e.g., tumors/cancers) including singular/plural forms, standalone mentions, and recognized inheritance patterns when clinically relevant
- Genetic conditions referenced by full names or clinically established abbreviations/shorthand (e.g., "ALD" for adrenoleukodystrophy), including those implied through inheritance patterns (e.g., "autosomal dominant disorder") or gene symbols directly representing conditions (e.g., "VHL" for Von Hippel-Lindau disease)
- Compound terms combining anatomical locations, clinical descriptors, symptomatic manifestations, or hyphens with medical conditions (e.g., "desmoid tumor", "pituitary-adrenal abnormality", "adenomatous polyps of the colon")
- Multi-word expressions representing recognized conditions (e.g., "Lesch-Nyhan syndrome"), their standard abbreviations (e.g., "L-N"), and clinically significant outcomes/complications (e.g., "sudden death")

Exclude standalone genes/proteins unless:
1) Integral to a condition's formal name (e.g., "APC" in "APC-associated polyposis"), or
2) Clinically recognized as direct shorthand for a condition (e.g., "WAS" for Wiskott-Aldrich syndrome).

Include all pathological state variants - modified (e.g., "benign tumors"), unmodified (e.g., "tumors"), or descriptive (e.g., "deficiency", "abnormality") - when contextually referring to medical conditions. |

Table 19: Initial task prompt and final optimized prompt from GRACE on task NCBI.

This is a biomedical sentence similarity task. Please carefully read the following sentences and rate the similarity of two input sentences. Choose between 'not similar', 'somewhat similar' and 'similar'.

Compare two biomedical sentences and classify their similarity based on shared key elements (entities, mechanisms, outcomes):

1. **Similar**: All key elements (entities, mechanisms, outcomes) are explicitly identical in both sentences, with no differences in specificity, scope, or implied relationships. Supplementary details (e.g., additional context or examples) that do not alter the core elements' identity or interpretation are permissible. Outcomes must match in both scope (e.g., global vs. localized) and specificity.

2. **Somewhat similar**: Share at least one concrete key element (entity, mechanism, or outcome) but differ in others. Differences include:
- Partial overlaps in elements
- Additional or omitted key elements*
- Variations in specificity (e.g., general vs. specific entities or mechanisms)
- Contextual differences affecting interpretation
- Outcomes differing in scope or specificity

3. **Not similar**: No concrete overlap in any key elements. Shared general themes (e.g., "cancer" or "cell death") without specific shared entities, mechanisms, or outcomes.

Respond strictly with "similar", "somewhat similar", or "not similar".

Table 20: Initial task prompt and final optimized prompt from GRACE on task Biosses.

Please use your domain knowledge in medical area to solve the questions.

Apply your medical expertise to systematically analyze clinical history, symptoms, diagnostic findings, and risk factors. Prioritize differential diagnoses by evaluating key distinguishing features, pathophysiological mechanisms, and complications most consistent with the presentation while distinguishing between primary etiologies and secondary associations.

Table 21: Initial task prompt and final optimized prompt from GRACE on task MedQA.

Given the text, choose between 'subjective' and 'objective'.

Classify the text as **subjective** (author's personal opinions) or **objective** (grounded in narrative/genre context).

**Subjective**: Direct critiques, evaluative language (e.g., "generic," "effective"), or claims assessing the work's quality, impact, or creative approach without narrative basis. Includes statements about the work's effect on the audience (e.g., "shows us," "makes it recognizable") when lacking narrative grounding, and assessments of the works strategy or execution framed as inherent attributes.

**Objective**: Descriptions tied to characters' perspectives (including their emotions, judgments, or rhetorical questions in dialogue/internal thoughts), plot, genre conventions, symbolism, hypotheticals, or structural elements within the work's internal logic. Evaluative terms remain objective only when explicitly describing narrative content (e.g., a "poignant" character moment) or genre-specific mechanisms.

**Key**: Prioritize context. Neutral terms become subjective if evaluating the work (e.g., "innovative approach," "operates by its own rules"). Emotional language or rhetorical questions are objective when tied to narrative context. Distinguish rigorously between external critique (subjective) and narrative-driven analysis (objective). Claims about audience impact require explicit reference to narrative mechanisms (e.g., "the protagonists isolation makes viewers uneasy") to be objective. Descriptions of a works tone or themes as inherent qualities ("bittersweet drama") are subjective unless anchored to specific narrative elements.

Table 22: Initial task prompt and final optimized prompt from GRACE on task Subj.

Tag the text according to the primary topic of the question. Choose from (A) Abbreviation, (B) Entity, (C) Description and abstract concept, (D) Human being, (E) Location, (F) Numeric value

Classify the answer type required for each question using one category:

**A** - Abbreviation (requires acronym/short form)

**B** - Entity (specific non-human terms, objects, procedures, attributes, or lists; excludes humans, locations, and human-established organizations)

**C** - Description/Concept (explanations, definitions, causes; no specific entities needed)

**D** - Human (requires explicit personal/group names, including human-established organizations such as companies, institutions, or groups)

**E** - Location (geopolitical regions, physical places such as buildings, landmarks, or natural features; digital environments such as URLs; nationality)

**F** - Numeric (values/counts unless part of an Entity's attributes)

**Guidelines**:

1. Classify based on the **answer's required information**, not the question's subject.

2. Prefer **C** over **D** unless the answer requires a specific personal/organizational name. For example:
- Use **C** for descriptions of roles, services, or achievements (e.g., "What does Company X specialize in?" → description of services).
- Use **D** only when a specific name is explicitly needed (e.g., "Which organization developed Product Y?" → organization name).

3. **B** applies when answers require specific terms (e.g., "What is X?" where the answer is X's name/acronym, such as a technical term or entity name) **or lists of non-human entities**. Use **C** for explanations/definitions even if X is a named entity.
- Example distinction:
- "What is the fear of hell called?" → **B** (term *stygiophobia*).
- "What does a chiropodist treat?" → **B** (specific terms like *feet, corns*).
- "What is the fear of hell?" → **C** (explanation of the phobia).
- "What are stars primarily composed of?" → **B** (specific terms like *hydrogen, helium*).
- Lists of attributes (e.g., "What features distinguish X?") use **B** if the answer requires specific terms (e.g., *tusks, ears*); use **C** if it requires explanations (e.g., "larger ears for thermoregulation").
- **Even if the question asks for a cause, effect, explanation, or definition**, use **B** if the answer is a specific term (e.g., "What turns litmus paper red?" → *acid*; "What is the term for the fear of heights?" → *acrophobia*).

4. **E** applies to locations/nationality even when tied to humans (e.g., "Where was Person Z born?" → **E**). Physical places include buildings/infrastructure regardless of organizational association. **Digital environments such as URLs or web addresses are also classified under E** (e.g., "What is the website for Organization X?" → **E**).

5. Attributes of entities (human-associated or otherwise) use their respective category (**B**, **E**, etc.). **Names of organizations/institutions always use D**, even when describing their type (e.g., "What kind of company is X?" → **D** if the answer is the organization's name; use **C** only for descriptive explanations unrelated to naming).

6. **F** applies **only** to standalone numeric values (e.g., "How many...?"). If a numeric is an attribute of an entity (e.g., temperature in a recipe, population count of a city), use the entity's category (**B**/**E**).

Table 23: Initial task prompt and final optimized prompt from GRACE on task TREC.

Read carefully the following premise and hypothesis, and determine the relationship between them. Choose from 'contradiction', 'neutral' and 'entailment'.

Classify if the premise entails, contradicts, or is neutral to the hypothesis. Focus on the core meaning, ignoring minor grammatical differences and pronoun changes that refer to the same entity. Carefully analyze negations to determine if they directly oppose the hypothesis, distinguishing between factual negations and those within personal opinions. Consider implied stances in rhetorical questions or challenges only when context provides strong evidence for the intended stance. Differentiate between opinions (e.g., beliefs, likelihoods) and factual assertions, ensuring opinions do not directly contradict hypotheses unless explicitly stated as factual claims. Answer with only 'entailment', 'contradiction', or 'neutral'.

Table 24: Initial task prompt and final optimized prompt from GRACE on task CB.

Name geometric shapes from their SVG paths.

Classify SVG paths by:

1. **Sides**:
- Count each 'L' command as one side, **including those that close the path**.
- **Closure**:
- **Entire path closure check first**:
- If the path's first and last points match, the entire path is closed. Sum **all** 'L' commands in the entire path as sides, **treating the path as a single continuous sequence and disregarding any intermediate 'M' commands. Do not split into subpaths in this case**.
- **Subpaths only if entire path is unclosed**:
- If the entire path is not closed, split it into subpaths at each 'M' command. For each subpath, sum its 'L' commands **only if** the subpath's first and last points match.

2. **Arc shapes** (prioritize if arcs exist):
- **Circle/Ellipse**: Closed path using **only** arcs (circle: equal radii; ellipse: unequal radii).
- **Sector**: Two lines from a shared vertex connected by an arc between their endpoints (arc must be present).

3. **Polygons** (no arcs):
- **Quadrilaterals** (4 sides):
- **Rectangle**: Four angles 90° (±5°) **in sequence**, with **opposite** sides (1st vs 3rd, 2nd vs 4th) <=5% length difference. **Prioritize over kite when criteria conflict**.
- **Kite**: **Two distinct pairs of adjacent sides** (each pair consecutive in path order) with <=5% difference — classify only if rectangle criteria are unmet.
- **Other**: Label as triangle, pentagon, etc., based on total sides.

**Tolerances**: 5% length difference; ±5° angle deviation. Verify side adjacency follows path order **strictly**.

Table 25: Initial task prompt and final optimized prompt from GRACE on task Geometry Shapes.

Detect the type of error in an English translation of a German source sentence.

Identify translation errors by comparing the German source and English translation. Classify errors into the **most specific applicable category**:

1. **Named Entities**: Incorrect translation of proper names, specific locations, organizations, or other unique entities (e.g., changing "Berlin" to "Munich"). Excludes adjective-based descriptors and administrative terms unless integral to the official name.
2. **Numerical Values**: Altered dates, numbers, units, or their **omissions** (e.g., "fourth" → "fifth", dropping "July 19 to August 3").
3. **Modifiers/Adjectives**: Omitted or changed descriptors (e.g., nationality, origin, material) that qualify a noun, excluding antonym substitutions. Includes regional/organizational adjectives not part of official names and **omissions of adjective phrases** (e.g., dropping "immovable architectural").
4. **Negation/Antonyms**: Added/removed negation or substituted **direct linguistic antonyms** (e.g., "can" → "cannot", "upper" → "lower"). Excludes contextual/conceptual opposites and directional/spatial antonyms (e.g., "north"→"south") that alter factual meaning.
5. **Facts**: Factual inaccuracies not covered by more specific categories, including **attribute changes** (e.g., color, role), mistranslations of administrative/geographical terms (e.g., "district" → "state"), and directional/spatial antonym errors affecting factual positions.
6. **Dropped Content**: Essential **clauses or full phrases** omitted (excluding numbers/dates/modifiers).

**Prioritization Order**:
1. Negation/Antonyms > All other categories when direct antonyms/negation are involved
2. Numerical Values (including date/number omissions) > Dropped Content
3. Modifiers/Adjectives > Facts when applicable
4. Named Entities > Facts unless descriptor errors apply

**Key Clarifications**:
- Administrative/geographical terms are Named Entities **only** if part of an official proper name. Type changes (e.g., district→state) fall under Facts.
- Omissions of dates/numbers always prioritize Numerical Values over Dropped Content.
- **Direct antonym substitutions within proper names** (e.g., "Lower Austria"→"Upper Austria") are classified under Negation/Antonyms.
- Conceptual opposites (e.g., "victim"→"victor") and directional/spatial antonyms (e.g., "north"→"south") that create factual distortions belong to Facts.
- Color/role changes and similar factual attribute errors fall under Facts unless covered by more specific categories.

Table 26: Initial task prompt and final optimized prompt from GRACE on task Salient Translation Error Detection.

Determine which of two sentences is sarcastic.

Identify sarcastic sentences by detecting contradictions between literal meaning and contextual intent. Analyze irony, exaggerated/dismissive language, rhetorical questions, and incongruities with common knowledge, situational context, or the speaker's expected perspective (prioritizing typical assumptions about the speaker if unspecified). Prioritize mismatches between stated sentiment (positive/negative) and contextual plausibility, including mock endorsement of implausible perspectives, obvious falsehoods, trivialization of significant issues, or alignment with viewpoints the speaker would obviously oppose given the context. Consider both overt contradictions and subtle incongruities, particularly when tone or intensity is disproportionately exaggerated relative to the situation's practical reality or the speaker's implied stance. Additionally, evaluate whether the statement critiques or mockingly endorses widely recognized frustrations, overhyped trends (regardless of their actual merit), or common societal critiques, as sarcasm often arises from these contexts. Pay special attention to rhetorical questions that ironically affirm or deny propositions based on prevailing attitudes, and to statements where the speaker's true stance is evident through contextual cues that contradict the literal message.

Table 27: Initial task prompt and final optimized prompt from GRACE on task Snarks.

Recommend movies similar to the given list of movies.

Recommend films similar to a provided list by analyzing genre, theme, era, target audience, critical reception, commercial success, and narrative elements. Follow these priorities:

1. **Era**: Prioritize era alignment only if a strong majority (>=75%) of the input films share a cohesive timeframe (e.g., same decade or within a 10-year period). When era is prioritized, recommendations must originate from the same timeframe unless no viable high-impact options exist.

2. **Impact & Appeal**: Favor films with comparable or greater critical/commercial success, particularly those with major cultural influence, awards recognition, or enduring audience resonance. Cultural influence includes genre-defining works, **parodies/satires with widespread recognition**, and films that set new standards within their categories. Prioritize this criterion over thematic alignment unless thematic connections are strongly supported by **specific narrative/stylistic evidence**. When era is prioritized, first select the highest-impact films within that era **regardless of genre mismatches**, unless explicit and substantial thematic connections justify an alternative choice.

3. **Themes & Narrative**: If era is inconsistent, focus on **concrete** thematic/narrative parallels (e.g., shared plot structures, directorial techniques, or character archetypes) validated by examples. Avoid relying on broad thematic concepts (e.g., "redemption," "identity") without specific narrative devices. Only use thematic alignment to override impact considerations when parallels are explicit, substantial, and directly tied to the input filmścore narrative/stylistic traits.

4. **Artistic Identity**: Highlight films with groundbreaking technical/artistic achievements, emotionally resonant storytelling, or **genre-redefining approaches that created new categories**, even when surface-level mismatches exist.

**Reconciliation Rules**:
- When era is prioritized, **exhaustively evaluate all era-aligned options** for impact before considering films from other timeframes. **Do not bypass era-aligned films unless they lack minimum viability (e.g., critical/commercial failure)**.
- Avoid abstract thematic links; require direct connections to the input filmścore traits (e.g., "shared use of nonlinear storytelling" vs. "both explore redemption").
- When input films span multiple genres, prioritize recommendations that excel in impact or innovation within any represented genre.
- When era-aligned options lack sufficient impact, consider high-impact films from other eras **only if they demonstrate definitive artistic/thematic DNA or genre-redefining status** with the input list.
- **Explicitly prioritize cultural impact over partial era/genre matches when the candidate film redefined its genre or achieved supreme critical/commercial dominance**.
- If thematic ties are weak or speculative, default to superior impact/artistic merit **even if this creates era/genre mismatches**.
- **Never use minor thematic overlaps to override era-aligned high-impact films when era is prioritized**.

Table 28: Initial task prompt and final optimized prompt from GRACE on task Movie Recommendation.

| Determine whether one sentence entails the next. |
| --- |

Determine if the premise logically entails the hypothesis. Apply these rules:

1. **Factive verbs** (e.g., "knows," "remembers"): Treat their complements as true in all contexts where they appear.
2. **Embedded factives**: If a factive verb's complement is embedded under any attitude (e.g., "believes," "suspects") **at any depth**, the attitude holder's mental state includes commitment to the complement's truth. This commitment propagates upward through all embedding attitudes but **does not imply the truth of clauses containing the factive verb itself** (e.g., in "A believes B knows C," A is committed to C's truth, not to B's knowledge of C).
3. **Non-factive attitudes**: Remain non-committal toward clauses lacking embedded factives, unless modified by Rule 2. Nested non-factive structures (e.g., "A assumes B thinks C") do not transfer commitments across attitude holders — only the innermost factive complement (if present) propagates truth commitments upward.

Assess entailment by checking if all committed truths (from factives/embedded factives) and explicit premise content necessarily imply the hypothesis through:
- **Semantic equivalence or hyponymy** (including generalization from specific terms to their hypernyms or contextually inferred roles)
- **Common-sense inferences** based on inherent and necessary categorical relationships (e.g., "attire" implies "clothing"; "overlooking from a cliff" implies elevation)
- **Lexical entailments** preserving truth conditions
- **Structural consistency**: The attitude holder (subject of the attitude verb) must remain identical between premise and hypothesis unless a hypernym or coreferential relationship exists. Changes to attitude holders without semantic justification invalidate entailment.

Table 29: Initial task prompt and final optimized prompt from GRACE on task Epistemic.

