# OpenReview forum: "No Loss, No Gain: Gated Refinement and Adaptive Compression for Prompt Optimization"
_NeurIPS.cc/2025/Conference — NeurIPS 2025 poster_

### Official Review · Reviewer_NhAd · 2025-06-25

**Clarity:** 3
**Significance:** 3
**Originality:** 3
**Rating:** 4
**Confidence:** 4

**Summary:**

This paper presents GRACE, an efficient framework for automatic prompt optimization in large language models (LLMs), addressing the challenges of instability and local optima in existing methods. GRACE integrates two synergistic strategies: gated refinement, which uses a feedback regulation gate and an update rejection gate to stabilize prompt updates by refining signals from both successful and failed samples, and adaptive compression, which distills the prompt’s core concepts to escape local optima when optimization stagnates. By strategically introducing information loss through these mechanisms, GRACE balances local refinement and global restructuring, enabling more effective exploration-exploitation in prompt space.

**Questions:**

1. Hopefully, the authors will add the relevant experiments mentioned in section Strengths And Weaknesses.

2. The author's idea is very interesting, which can be compared to the optimization of a neural network, and the optimization process is made more stable by gradient clipping and batch size. Then my problem also stems from this, if I reduce the number of parts or tokens modified each time, so as to achieve a more stable optimization similar to the neural network optimization to reduce the learning rate. In addition, if a small portion of the modification is returned to the original prompt after each optimization, it is similar to the Dropout of neural network optimization.

**Ethical Concerns:**

["NO or VERY MINOR ethics concerns only"]

**Final Justification:**

Thanks to the reviewer for the response, this addressed my concerns about this work, and I will keep my score.

**Limitations:**

yes

**Quality:**

3

**Strengths And Weaknesses:**

Strengths
1. High Efficiency and Performance Gains: GRACE demonstrates significant performance improvements (4.7% on BBH, 4.4% on domain-specific tasks, and 2.7% on general NLP tasks) while requiring only 25% of the prompt generation budget compared to prior methods. This efficiency is achieved through its synergistic strategies of gated refinement and adaptive compression, which stabilize updates and escape local optima effectively.
2. Innovative Dual-Strategy Design: The framework integrates a feedback regulation gate and update rejection gate to refine update signals, alongside adaptive compression to distill core concepts during stagnation. This design balances local refinement and global restructuring, addressing both instability and local optima in prompt optimization.
3. Rigorous Experimental Validation: Extensive experiments across 11 tasks in diverse domains and ablation studies (e.g., on feedback ratios and compression triggers) validate GRACE’s robustness and component effectiveness, providing strong empirical support for its claims.

Weaknesses
1. This paper lacks some relevant comparative experiments to verify the effectiveness and scalability of the proposed method. For example, can optimized prompts be used across models, and does the use of models of different sizes and architectures as prompt generators have an impact on the final performance?

---

> ### Author Rebuttal · Authors · 2025-07-30
>
> We thank the reviewer for the encouraging feedback and inspiring questions.
>
> ### ***Q1: More Comparative Experiments***
>
> We fully agree with your suggestions.
> In Appendix Table 6, we already report results across different base LLMs. Motivated by your suggestion, we further conduct two additional experiments: (1) evaluating the **transferability of optimized prompts** across base LLMs, and (2) analyzing the **impact of different optimizer LLMs**. Both experiments will be included in the appendix of the revised version.
>
> **(1) Transferability of Optimized Prompts Across Base LLMs**
>
> **Our GRACE-optimized prompts exhibit a degree of generalizability but are most effective when applied to the same base LLM.**
>
> Since different base LLMs vary in architecture, pretraining data, and instruction tuning, we evaluate whether prompts optimized for one model generalize to others. Specifically, we optimize prompts using **DeepSeek-V3-0324** as the target base LLM on five BBH tasks, then evaluate them on **Llama-3.3-70B-Instruct** and **GPT-4.1** without further tuning.
>
> | | | **DeepSeek-V3** | | | | | **Llama-3.3-70B** | | | | | **GPT-4.1** | |
> | :--- | :---: | :---: | :---: | :---: | :---: | :---: | :---: | :---: | :---: | :---: | :---: | :---: | :---: |
> | **Tasks** | **Task (ZS)** | **PromptAgent** | **GRACE** | | | **Task (ZS)** | **PromptAgent** | **GRACE** | | | **Task (ZS)** | **PromptAgent** | **GRACE** |
> | Geometry   |62.50 | 88.50 | **97.00**  | | |72.50  |70.00  |**75.50** | | |43.00  |50.00  |**52.00**  |
> | Translation |65.71 | 77.86 |**84.29** | | |70.00  |70.71  |**72.86** | | |72.86  |77.14  |**80.71**  |
> | Snarks       |85.26 | 92.63 | **94.74** | | |**92.63**  |**92.63**  |**92.63**  | | |**93.68**  |89.47  |**93.68**  |
> | Movie        |78.57 | 92.63 | **97.14** | | |69.29  |80.00  |**92.86**  | | |75.00  |82.86  |**92.14**  |
> | Epistemic  |92.20 | 95.50 | **97.50** | | |85.25  |87.00  |**92.00**  | | |**87.00**  |84.25  |82.50 |
> | Average     |77.45 | 89.42 | **94.13** | | |77.93  |80.07  |**85.17**  | | |74.31  |76.74  |**80.21**  |
>
> Task (ZS) refers to the task's initial prompt. PromptAgent is a strong automatic prompt optimization bsaeline, and GRACE is our method.
>
> In most cases, GRACE-optimized prompts outperform both the initial prompts and those optimized by PromptAgent, indicating their generalizability.
> However, the performance gains are highest on the model used for optimization (DeepSeek-V3) and generally smaller when transferred to other models. In a few instances (e.g., Epistemic on GPT-4.1), the optimized prompts even underperform compared to the initial ones. These results suggest that while GRACE prompts exhibit partial transferability, the optimized prompt is most effective when applied to the target base LLM.
>
> **(2) Impact of Different Optimizer LLMs**
>
> **Frontier 'Thinking LLMs' are well-suited as the optimizer LLMs, and models with weaker analytical and reasoning abilities may struggle to effectively optimize prompts on some tasks.**
>
> Since the optimizer LLM plays a central role in our method, we assess how its ability affects performance. Besides our default optimizer (**DeepSeek-R1**), we experiment with another frontier thinking model (**o3**) and a general-purpose instruction-tuned model (**Llama-3.3-70B-Instruct**):
>
> | | | **DeepSeek-R1** | | | | |  **o3** | | | | | **Llama-3.3-70B** | |
> | :--- | :---: | :---: | :---: | :---: | :---: | :---: | :---: | :---: | :---: | :---: | :---: | :---: | :---: |
> | **Tasks** | **Task (ZS)** | **PromptAgent** | **GRACE** | | | **Task (ZS)** | **PromptAgent** | **GRACE** | | | **Task (ZS)** | **PromptAgent** | **GRACE** |
> | Geometry   |62.50 | 88.50 | **97.00**  | | |62.50 | 86.00 | **97.00**  | | |62.50 | 76.50 | **81.00**  |
> | Translation |65.71 | 77.86  |**84.29** | | |65.71 | 77.14 |**85.00** | | |65.71 | 78.57 |**83.57** |
> | Snarks       |85.26 | 92.63 | **94.74** | | |85.26 | 90.53 | **93.68** | | |85.26 | 90.53 | **94.74** |
> | Movie        |78.57 | 92.63 | **97.14** | | |78.57 | 94.29 | **97.14** | | |78.57 | 93.57 | **95.71** |
> | Epistemic  |92.20 | 95.50 | **97.50** | | |92.20 | **97.00** | **97.00** | | |92.20 | 94.00 | **95.00** |
> | Average     |77.45 | 89.42 | **94.13** | | |77.45 | 88.99 | **93.96** | | |77.45 | 86.63 | **90.00** |
>
>
> We observe three key findings:
>
> 1. GRACE consistently outperforms baselines across all settings, demonstrating its robustness and effectiveness regardless of the optimizer model used.
>
> 2. DeepSeek-R1 and o3 yield comparable results, suggesting that frontier models with strong reasoning capabilities are both viable and effective as optimizer LLMs.
>
> 3. When using Llama-3.3-70B-Instruct as the optimizer, GRACE and PromptAgent achieve performance comparable to that with other optimizers on most tasks. However, on some tasks such as Geometry, its performance is substantially worse, indicating that it fails to sufficiently optimize the prompt.
>
> These results support our recommendation that the optimizer LLM be selected from models with strong analytical and reasoning abilities.
>
> ### ***Q2: Integrating Techniques from Neural Network Optimization***
>
> Thank you for your insightful suggestions.
> Your analogy between prompt optimization and neural network training is closely aligned with the motivation behind our GRACE method. **It has great potential to stabilize the prompt optimization process, but the practical applicability needs to be verified through empirical experiments**.
>
> Our work identifies a key issue in existing reflection-based prompt optimization methods: unregulated updates often lead to instability and convergence to suboptimal prompts.
> To address this, we draw a parallel between optimizing prompts based on error samples and gradient descent, and introduce three regularization-inspired designs: leveraging correct samples, selectively accepting updates, and adaptive compression.
> We agree that additional techniques from network training—such as limiting the magnitude of prompt edits (akin to lowering the learning rate), or reintroducing parts of the original prompt (akin to dropout)—may further enhance stability.
> Intuitively, these strategies could mitigate the issue of overly aggressive updates and help smooth the optimization trajectory.
> However, their effectiveness in the context of prompt optimization remains an open question and would require empirical validation.
> In our work, we experimented with several training-inspired strategies (e.g., momentum, adaptive learning rate), but found that only a subset transferred effectively.
> **Many techniques, while conceptually appealing, are difficult to apply in practice due to the discrete nature of prompt space and the non-deterministic behavior of optimizer LLMs, which can instead introduce additional noise and instability.**
> The current design of GRACE reflects the outcome of extensive empirical investigation and tuning.
>
> We greatly appreciate your suggestions and plan to continue explore how optimization theory and techniques from neural network training can be systematically adapted to prompt optimization in future work.

---

> > ### Comment · Reviewer_NhAd · 2025-08-08
> >
> > Thanks to the reviewer for the response, this addressed my concerns about this work, and I will keep my score, it is already positive.

---

### Official Review · Reviewer_scUS · 2025-07-02

**Clarity:** 3
**Significance:** 2
**Originality:** 3
**Rating:** 4
**Confidence:** 3

**Summary:**

This paper considers the problem of automatically optimizing prompts. The key innovation lies in two synergistic strategies: 1) Gated Refinement, which filters prompt updates via feedback regulation and update rejection gates to prevent unstable or overly aggressive updates that could lead to overfitting errors, and 2) Adaptive Compression, which restructures prompts by distilling core concepts when optimization stagnates. These strategies introduce controlled information loss to promote stable updates and escape local optima. The authors tested the proposed method on 11 tasks across three domains, showing improved performance and reduced prompt budget.

**Questions:**

see the last two paragraphs in the section of Strengths And Weaknesses*

**Ethical Concerns:**

["NO or VERY MINOR ethics concerns only"]

**Limitations:**

the authors adequately addressed the limitations and potential negative societal impact of their work

**Quality:**

3

**Strengths And Weaknesses:**

The proposed method is conceptually well-motivated. By combining the mechanisms of gated refinement and adaptive compression, it equipped a practical balance between exploration and exploitation in the prompt space. The authors relate this idea to the information bottleneck principle, providing theoretical insights into the advantages of controlled information loss.

Experiments show that the proposed method outperforms existing methods consistently across the NLP benchmark tasks. Gains are also seen in challenging settings such as BBH. Its performance remains consistent across different LLMs, showing its robustness and generalisability. Furthermore, reducing the API cost of the proposed method is essential for its practical use.

The approach appears to rely on carefully crafted meta-prompts, as shown in Table 11 and Table 12. However, there is no analysis of the sensitivity of the proposed method to these templates. As meta-prompt design affects how the optimiser LLM refines prompts, this could be a hidden factor influencing performance.

In the experiments, the tasks are mainly limited to classification tasks and reasoning benchmarks. While this is impressive, it does not conduct prompt optimization for tasks such as summarisation, question answering or instruction following. This restricts the scope of the claimed generalizability.

---

> ### Author Rebuttal · Authors · 2025-07-30
>
> We thank the reviewer for the insightful feedbacks. In the revised version, we will include the following experiments and analysis in the appendix.
>
> ### ***Q1: Sensitivity to Meta-Prompt Design***
>
> **We conduct sensitivity experiments by paraphrasing each key section of the meta-prompts, demonstrating the robustness of our GRACE method to variations in meta-prompt design.**
>
> Given the known sensitivity of LLMs to prompts, the design of meta-prompts can affect the effectiveness of prompt optimization.
> In constructing the meta-prompts, we first clarify the intended role of the optimizer LLM, then express it through clear, modular instructions.
> To evaluate the robustness of our method to meta-prompts, we paraphrase each key section of the meta-prompts used for **optimization** in Tabel 11 and **compression** in Table 12.
> For each section, we create 5 paraphrased variants and evaluate performance on 5 tasks from the BBH benchmark, reporting the mean and 95% confidence interval per task.
>
> (1) The meta-prompt for **optimization** consists of the following four components:
>
> * **Main Task (MT)**: Your task is to optimize the current prompt for a language model performing a specific task. The goal is to correct previously failed predictions while preserving the model’s correct behavior on already successful examples.
> * **Preserve Correctness (PC)**: Ensure the model, instructed by the optimized prompt, continues to predict correct answers for all successful examples. In addition to prediction correctness, maintain the model’s original correct solutions and response for these cases as much as possible.
> * **Refine to Fix Errors (FE)**: For failed examples, attempt to correct them by refining the prompt’s instructions — for example, by adding clearer or more complete guidance. Any new content should integrate naturally with the current prompt and form a coherent task instruction. Avoid special-case logic, examples, or instructions targeted at individual cases.
> * **Additional Guidelines (AG)**: - Prompt modifications should always aim to preserve model's correct behavior on successful examples. ...
>
> Performance with paraphrased versions of each component is shown below:
>
> $$
> \\begin{array}{l l | c c c c c c}
> & \text{Method} & \text{Geometry} & \text{Translation} & \text{Snarks} & \text{Movie} & \text{Epistemic} & \text{Avg.} \\\\
> \\hline
> & \\text{GRACE}       & 97.00 & 84.29 & 94.74 & 97.14 & 97.50 & 94.13 \\\\
> & \\text{MT}       & 97.00 (0.71) & 84.29 (1.51) & 94.32 (1.60) & 97.14 (0.51) & 97.45 (0.37) & 94.04 \\\\
> & \\text{PC}       & 96.40 (1.39) & 84.57 (1.26) & 94.95 (0.47) & 96.86 (0.39) & 97.65 (0.29) & 94.09 \\\\
> & \\text{FE}       & 97.00 (1.00) & 83.57 (1.34) & 94.95 (1.15) & 96.86 (0.39) & 97.35 (0.60) & 93.95 \\\\
> & \\text{AG}       & 96.90 (0.74) & 84.00 (1.40) & 94.32 (1.49) & 97.14 (0.51) & 97.25 (0.40) & 93.92 \\\\
> \\end{array}
> $$
>
> (2) The meta-prompt for **compression** contains two main components:
>
> * **Main Task (MT)**: The prompt may have accumulated redundant, overly specific, or ineffective wording across previous iterations. Your goal is to ...
> * **Additional Guidelines (AG)**: Eliminate instructions that are verbose, ambiguous, or unlikely to generalize ...
>
> Performance with paraphrased compression prompts is shown below:
>
> $$
> \\begin{array}{l l | c c c c c c}
> & \text{Method} & \text{Geometry} & \text{Translation} & \text{Snarks} & \text{Movie} & \text{Epistemic} & \text{Avg.} \\\\
> \\hline
> & \\text{GRACE}       & 97.00 & 84.29 & 94.74 & 97.14 & 97.50 & 94.13 \\\\
> & \\text{MT}       & 97.00 (0.50) & 84.14 (1.28) & 94.53 (1.56) & 97.00 (0.78) & 97.40 (0.52) & 94.01 \\\\
> & \\text{AG}       & 97.29 (1.64) & 84.00 (0.96) & 94.74 (1.49) & 97.28 (0.32) & 97.55 (0.33) & 94.17 \\\\
> \\end{array}
> $$
>
> Across all paraphrased variants, performance remains highly stable with minimal variance, demonstrating our method's strong robustness to variations in meta-prompts.
> In addition, this finding aligns with our observations in search-based prompt optimization methods, where capable LLMs are often insensitive to minor phrasing variations when the core intent is preserved.
>
> ### ***Q2: Additional Experimental Tasks***
>
> **We conduct experiments on two widely-used summarization tasks, where GRACE also demonstrates significant performance improvements over other methods.**
>
> Our original experiments demonstrate the effectiveness of our method on complex reasoning, domain-specific, and natural language understanding tasks, which are the most common and standard benchmark tasks for automatic prompt optimization methods.
> Notably, tasks such as BBH and MedQA can be categorized as question answering, as they involve answering explicit questions based on given options.
> To further demonstrate the broader applicability of our method, we evaluate it on two summarization tasks: the Reddit TIFU dataset [1] and the Amazon Fine Food Reviews dataset [2]. Task instructions are sourced from Super-NaturalInstructions [3], and performance is measured using Rouge-L.
>
> $$
> \\begin{array}{l l | c c c}
> & \\text{Method} & \\text{Reddit} & \\text{Amazon} & \\text{Avg.} \\\\
> \\hline
> & \\text{Task (ZS)}       & 12.21 & 16.78 & 14.50 \\\\
> & \\text{Task (FS)}       & 12.64 & 18.52 & 15.58 \\\\
> \\hline
> & \\text{EvoPrompt}      & 13.24 & 19.24 & 16.24 \\\\
> & \\text{OPRO}           & 13.13 & 20.35 & 16.74 \\\\
> & \\text{APO}            & 14.33 & 21.45 & 17.89 \\\\
> & \\text{PromptAgent}      & 16.29 & 21.29 & 18.79 \\\\
> & \\text{GRACE}          & \textbf{17.60} & \textbf{23.76} & \textbf{20.68} \\\\
> \\end{array}
> $$
>
> Task (ZS) is the initial prompt. As shown, GRACE achieves the highest performance across both datasets, demonstrating its effectiveness on summarization tasks and supporting its generalizability beyond classification and reasoning tasks.
>
> [1] Kim et al.; "Abstractive Summarization of Reddit Posts with Multi-level Memory Networks"; NAACL 2019.
>
> [2] https://www.kaggle.com/datasets/snap/amazon-fine-food-reviews
>
> [3] Wang et al.; "Super-NaturalInstructions: Generalization via Declarative Instructions on 1600+ NLP Tasks"; EMNLP 2022.

---

### Official Review · Reviewer_M4zQ · 2025-07-03

**Clarity:** 4
**Significance:** 3
**Originality:** 3
**Rating:** 5
**Confidence:** 4

**Summary:**

This paper studies the task of prompt optimization, which is to automatically improve a task-specific prompts for better performance. An iterative method of two major components is proposed. In Gated Refinement part, the method learns from known successful and failed cases to propose new prompts what are then selected based on performance on a development set. In Adaptive Compression, the method will compress a potentially biased prompts to get rid of too specific information. Experiments show the proposed method consistently achieves stronger performance compared with multiple baselines.

**Questions:**

The paper overall is clear to me and I have no further question. Please see the weaknesses above.

**Ethical Concerns:**

["NO or VERY MINOR ethics concerns only"]

**Final Justification:**

During rebuttal, my concerns are addressed and the additional results and discussion improve the quality of the paper. I have increased my rating.

**Limitations:**

Please see comment above.

**Quality:**

3

**Strengths And Weaknesses:**

### Strengths
- The idea of adaptive compression is interesting and the experiments clearly shows the advantage of including such a compression step during the iterations.
- The paper is clearly written and easy to read


### Weaknesses
- The experiments would benefit from more discussion on how the hyperparameters are selected. Some ablation studies about parameters are included but it does not seem enough for me to decide the parameters on a new task/dataset.
- I would expect a more comprehensive discussion on the limitations of proposed method with potential limitations on the generalizability of the method on different tasks and domains.

---

> ### Author Rebuttal · Authors · 2025-07-30
>
> Thank you for your valuable reviews and suggestions.
> In the revised version, we will include more ablation studies on hyperparameters and further clarify how the hyperparameters are determined in the appendix. Additionally, we will expand the limitation section to better define the scope of applicability of our method.
>
> ###  ***Q1: Hyper-Parameter Ablation and Selection***
>
> **The hyperparameters we select are based on extensive empirical results, balancing efficiency and performance, and are suitable for a wide range of tasks.**
> Our experimental results show that, compared to existing methods, our approach achieves significant performance improvements with much lower overhead.
>
> Our GRACE method involves three key hyper-parameters: (1) the number of correct and incorrect samples used per update, denoted as $\left|S_t'\right|$ and $\left|F_t'\right|$, (2) the maximum number of optimization steps $T$, and (3) the number of consecutive rejections $K$ required to trigger compression.
>
> (1) **We choose $\left|S_t'\right| = \left|F_t'\right| = 3$, as it provides the most effective update signal.** The comparison experiments and analysis can be found in Figures 5.a and 6.
>
> (2) **We choose $T = 80$, which provides optimal performance for most tasks, while achieving a balance between performance and cost.** Increasing $T$ generally improves performance, but the gains diminish. Additionally, computational cost and resource requirements grow linearly with $T$.  Below is the performance of GRACE with different values of $T$.
>
> $$
> \\begin{array}{l l | c c c c c c}
> & \text{Method} & \text{Geometry} & \text{Translation} & \text{Snarks} & \text{Movie} & \text{Epistemic} & \text{Avg.} \\\\
> \\hline
> & \\text{T=40}       & 91.00 & 82.96 & \textbf{94.74} & 92.86 & 91.25 & 90.56 \\\\
> & \\text{T=60}       & 93.50 & 83.71 & \textbf{94.74} & 94.29 & 95.70 & 92.39 \\\\
> & \\text{T=80}       & 97.00 & 84.29 & \textbf{94.74} & \textbf{97.14} & \textbf{97.50} & 94.13 \\\\
> & \\text{T=100}     & \textbf{97.50} & 84.29 & \textbf{94.74} & \textbf{97.14} & \textbf{97.50} & 94.23 \\\\
> & \\text{T=120}     & \textbf{97.50} & \textbf{85.00} & \textbf{94.74} & \textbf{97.14} & \textbf{97.50} & \textbf{94.38} \\\\
> & \\text{T=140}     & \textbf{97.50} & \textbf{85.00} & \textbf{94.74} & \textbf{97.14} & \textbf{97.50} & \textbf{94.38} \\\\
> \\end{array}
> $$
>
> As shown, $T=80$ achieves the best results on 3 tasks and near-optimal performance on the remaining 2. Further increasing $T$ yields minimal improvements but incurs a linear increase in cost.
> Thus, to balance performance and efficiency, we use $T=80$ as the default.
> Additionally, we observe that when no improvement is seen over 20 consecutive steps, further iterations rarely help.
> Therefore, on new tasks, we recommend setting $T = 80$, possibly combined with an early stopping criterion of 20 stagnant steps.
>
> (3) **We select $K = 5$, as it provides optimal performance across most tasks.**
>
> $$
> \\begin{array}{l l | c c c c c c}
> & \text{Method} & \text{Geometry} & \text{Translation} & \text{Snarks} & \text{Movie} & \text{Epistemic} & \text{Avg.} \\\\
> \\hline
> & \\text{K=3}       & 91.50 & 81.43 & 93.68 & 94.29 & 93.00 & 90.78 \\\\
> & \\text{K=4}       & 97.00 & \textbf{84.29} & \textbf{94.74} & \textbf{97.14} & 96.25 & 93.88 \\\\
> & \\text{K=5}       & 97.00 & \textbf{84.29} & \textbf{94.74} & \textbf{97.14} & \textbf{97.50} & \textbf{94.13} \\\\
> & \\text{K=6}       & \textbf{97.50} & 83.57 & \textbf{94.74} & 96.43 & 95.50 & 93.55 \\\\
> & \\text{K=7}       & 94.00 & 84.14 & \textbf{94.74} & \textbf{97.14} & 91.25 & 92.25 \\\\
> \\end{array}
> $$
>
> We observe that both smaller and larger values of $K$ lead to decreased performance.
> A small $K$ may trigger premature compression before sufficient optimization has been explored, destabilizing the optimization process.
> On the other hand, a large $K$ may provide more optimization opportunities but could also waste resources, reducing the number of effective optimization in a limited number of iterations, leading to lower final performance.
> Therefore, we suggest set $K = 5$ to achieve a balance between exploration and exploitation in the prompt optimization space.
>
> In summary, GRACE’s hyper-parameters are chosen to ensure strong and stable performance across diverse tasks, and we provide concrete recommendations (e.g., $\left|S_t'\right| = \left|F_t'\right| = 3$, $T=80$, $K=5$) for adapting to new datasets.
>
>
> ###  ***Q2: Limitations on Domain-Specialized Tasks***
>
> We have demonstrated that GRACE achieves strong performance on complex reasoning, natural language understanding, and domain-specific tasks, significantly surpassing existing methods in both efficiency and effectiveness.
> However, upon further analysis, we find that the **performance gains on tasks requiring specialized domain knowledge are relatively limited**. As shown in Table 1, the performance improvement on domain-specific tasks, particularly the MedQA dataset, is modest.
> Our failure case analysis reveals that certain samples in the MedQA dataset require highly specialized medical knowledge, which both the optimizer and base LLMs lack. This indicates that our method struggles to effectively address such cases.
> It is important to emphasize that this limitation is not unique to our method but reflects a gap in current automatic prompt optimization approaches, which all rely solely on the capabilities of the optimizer and base LLMs.
>
> Improving the generalizability and practicality of APO methods to better handle domain-specific knowledge is an important direction of our future research.
> To address this, we are exploring the integration of retrieval-augmented generation (RAG) methods. By enabling the optimizer LLMs to access domain-specific knowledge bases, we can incorporate missing domain knowledge into the prompts.

---

> > ### Comment · Reviewer_M4zQ · 2025-08-05
> >
> > Thank you for the detailed response and additional results! My concerns are resolved well and I would like to increase my overall rating. The additional results and discussion should be included in the revision.

---

> > > ### Author Response · Authors · 2025-08-05
> > >
> > > We are glad that our additional results address your concerns and sincerely appreciate your thoughtful reviews.
> > > We will incorporate the new results and discussion into the revised version of the paper as suggested.

---

### Official Review · Reviewer_DseC · 2025-07-09

**Clarity:** 4
**Significance:** 3
**Originality:** 3
**Rating:** 5
**Confidence:** 3

**Summary:**

This paper studies prompt optimization. The authors propose a two-stage gated refinement and adaptive compression to enable more efficient prompt optimization (GRACE). . The gated refinement strategy uses a feedback regulation gate and an update rejection gate to refine update signals to produce stable prompt improvements. The adaptive compression strategy distills the prompt’s core concepts, restructuring the optimization trace and opening new path when prompt optimization gets stuck. The authors prove that GRACE achieves better average relative performance improvements of over state-of-the-art methods, including multiple auto prompt optimization and manual prompt optimization.

**Questions:**

1. The authors listed the number of search size in the figure as well as Cost($) comparison of base and optimizer LLMs in Table 3. I'd like to see whether these two directly translate to the computing time of the algorithms.
2. Table 2 shows a very nice illustration of how GRACE works step by step. I'd like to see whether one can have similar tables for other baseline methods as well to better compare and understand how all these methods work and why GRACE is better.

**Ethical Concerns:**

["NO or VERY MINOR ethics concerns only"]

**Limitations:**

Yes

**Quality:**

3

**Strengths And Weaknesses:**

Strengths

This paper studies a very important problem of prompt optimization. The authors clearly convey the motivation of this work, the architecture of the proposed GRACE algorithm, how GRACE works with step-by-step example (e.g., Table 2), how and whether the component in GRACE works with detailed ablation study.

I also like the fact that GRACE conducts more targeted optimization, resulting in low cost on both the base and optimizer LLMs, and can achieve good performance with much lower overall costs. It appears to me that GRACE is able to push the overall frontier of Cost / Performance curve.

The insights on why Baseline methods are easier to get stuck in local optima are also helpful: They either perform small, random updates or make overly aggressive changes, both of which lead to unstable prompt updates. As a result,  the instability forces baseline methods to explore a large number of candidates at each step to find improved prompts.


Weaknesses
I have minor comments for the weaknesses.

1)The authors listed the number of search size in the figure as well as Cost($) comparison of base and optimizer LLMs in Table 3. I'd like to see whether these two directly translate to the computing time of the algorithms. 2) Table 2 shows a very nice illustration of how GRACE works step by step. I'd like to see whether one can have similar tables for other baseline methods as well to better compare and understand how all these methods work and why GRACE is better.

---

> ### Author Rebuttal · Authors · 2025-07-30
>
> We thank the reviewer for the encouraging feedback and incisive reviews.
> In the revised version, we will include the relevant experiments in the appendix.
>
> ### ***Q1: Computing Time***
>
> **The total computing time scales approximately linearly with the number of search size and the reported costs.**
>
> The total computing time can be decomposed into two components: (1) the time spent by the optimizer LLM to update prompts, and (2) the time spent by the base LLM to evaluate these prompts. Let $N$ denote the number of search sizes, and $T$ the total computing time. The approximate relationships are:
>
> $T_{EvoPrompt}=N_1 \cdot T_{opt1} + N_1 \cdot T_{base}$
>
> $T_{OPRO}=N_1 \cdot T_{opt2} + N_1 \cdot T_{base}$
>
> $T_{APO}= N_1 \cdot T_{opt2} \cdot 2 + N_1 \cdot T_{base} \cdot \alpha_{UCB}$
>
> $T_{PromptAgent}= N_1 \cdot T_{opt2} \cdot 2 + N_1 \cdot T_{base}$
>
> $T_{GRACE(ours)} =N_2 \cdot T_{opt2} + N_2 \cdot T_{base}$
>
> In the first half of the **optimization time**, $T_{\text{opt1}}$ refers to the time for lightweight optimization (e.g., paraphrasing), while $T_{\text{opt2}}$ involves more complex optimization using additional context such as failure cases or optimization history.
> APO and PromptAgent incur twice the optimization time, as they require generating suggestions followed by prompt updates. In contrast, GRACE directly optimizes prompts based on correct and incorrect examples, avoiding this overhead.
> Notably, the search sizes for GRACE, $N_2 = 80$, are significantly smaller than $N_1 \approx 300$ for other methods.
>
> In the second half of the **evaluation time**, $T_{\text{base}}$ denotes the time to evaluate a prompt on the validation set, which is nearly identical across methods.
> APO reduces evaluations by a factor of $\alpha_{\text{UCB}} \approx \frac{1}{3}$ via a UCB algorithm.
> In practice, we observe that $T_{\text{base}} > T_{\text{opt2}} > T_{\text{opt1}}$, meaning evaluation constitutes the main portion of overall computing time.
>
> Thus, computing time can be directly estimated from the number of search sizes, though the per-step compute varies across methods.
>
> **Likewise, total cost $C$ scales approximately linearly with the search size $N$ and follows a similar formulation.**
>
> $C_{EvoPrompt}=N_1 \cdot C_{opt1} + N_1 \cdot C_{base1}$
>
> $C_{OPRO}=N_1 \cdot C_{opt2} + N_1 \cdot C_{base1}$
>
> $C_{APO}= N_1 \cdot C_{opt2} \cdot 2 + N_1 \cdot C_{base2} \cdot \alpha_{UCB}$
>
> $C_{PromptAgent}= N_1 \cdot C_{opt2} \cdot 2 + N_1 \cdot C_{base2}$
>
> $C_{GRACE(ours)} =N_2 \cdot C_{opt2} + N_2 \cdot C_{base2}$
>
> Given approximately constant per-call cost and latency, we can estimate computing time as:
> **$T_{method} \approx C_{method} \cdot \frac{T_{opt} + T_{base}}{C_{opt} + C_{base}}$**
>
> In summary, although implementation details (e.g., API latency, meta-prompt design, hyperparameters) can affect the exact computing time and cost, the general trends and relationships hold. GRACE consistently achieves lower compute and cost primarily by substantially reducing the number of search sizes.
>
> ###  ***Q2: Illustrative Examples***
>
> To facilitate a clearer comparison, we will include illustrative examples of the prompt optimization process for OPRO (as a representative search-based method) and APO (as a representative reflection-based method).
>
> **Below is the process of OPRO on the CB task.**
>
> | **State** | **Prompt** | **Score** |
> | :--- | :--- | :--- |
> | Step 0 (Initial) | Read carefully the following premise and hypothesis, and determine the relationship between them. Choose from contradiction, neutral, or entailment. | 89.3 |
> | Step 1 (Parent 0) | Examine the premise for explicit statements or logical conclusions that directly support the hypothesis. Determine the relationship as contradiction, neutral, or entailment. | 89.3 |
> | **Step 4 (Parent 0)** | First, determine if the hypothesis is a logically necessary conclusion of the premise (entailment). If the premise directly negates or explicitly opposes the hypothesis, choose contradiction. If neither condition is definitively met, select neutral. | **91.1** |
> | Step 7 (Parent 4) | First, check if the premise definitively confirms the hypothesis (entailment). If the premise directly negates or makes the hypothesis logically impossible, select contradiction. If neither condition applies, choose neutral. | 89.3 |
> | Step 16 (Parent 4) | First, confirm if the hypothesis is explicitly stated or irrefutably entailed by the premise—select entailment. If the premise directly contradicts or logically invalidates the hypothesis, choose contradiction. If neither condition is definitively met, select neutral. | 91.1 |
> | Step 20 (Parent 4) | First, ascertain whether the hypothesis is explicitly stated or logically inescapable from the premise (entailment). If the premise explicitly negates the hypothesis or renders it logically invalid, select contradiction. When neither definitive confirmation nor definitive refutation exists without external inference, choose neutral. | 89.3 |
>
> OPRO generates 15 new candidates per step based on top-performing prompts.
> The prompts shown are the best examples at the selected steps. As observed, the generated candidates preserve the original semantics, resulting in limited diversity and slow improvements.
>
> **Below is the process of APO on the CB task.**
>
> | **State** | **Prompt** | **Score** |
> | :--- | :--- | :--- |
> | Step 0 (Initial) | Read carefully the following premise and hypothesis, and determine the relationship between them. Choose from contradiction, neutral, or entailment. | 89.3 |
> | **Step 1** (Parent 0) | Analyze the given premise and hypothesis step by step:\n1. Identify explicit statements, negations **(e.g., \'don’t\', \'never\')**, uncertain language **(e.g., \'might\', \'I think\')**, hypotheticals, or questions in the premise.\n2. Determine if the hypothesis is directly supported (entailment), directly opposed (contradiction), or neither (neutral) based on these elements.\n   - For contradictions: Check if the premise explicitly denies the hypothesis or expresses doubt about it **(e.g., \'I don’t think X\' contradicts \'X will happen\')**.\n   - For entailment: Verify the premise provides clear, unambiguous support for the hypothesis.\n   - For neutral: Apply if the premise is uncertain, asks a question without an answer, or lacks information to confirm/deny the hypothesis.\n3. Avoid assuming unstated intentions **(e.g., a question like \'Should they spend more?\' does not entail \'they should spend more\' unless explicitly agreed)**. \n\nRelationship options: contradiction, neutral, entailment. | **94.6** |
> | Step 2 (Parent 1) | Assess the connection between a premise and hypothesis using structured evaluation: \n\n1. Examine language and reasoning indicators:  \n   - **Direct assertions: Recognize overt claims, denials (e.g., "cannot"), or confirmations.**  \n   - **Indirect links: Uncover implied logic (e.g., modus ponens: given "If A, then B" and A, infer B), assumptions, or persuasive techniques (e.g., rhetorical questions suggesting answers).**  \n\n2. Determine the connection type:  \n   - Contradiction: ...**(e.g., "X is untrue" or "If X, then ¬Y" with X verified)**.  \n   - Entailment: ...(including via conditionals or rhetorical cues).  \n   - Neutral: ...  \n\n3. Protocols:  \n   - **Conditionals: Interpret "If X, then Y" as entailment if X is validated in the premise.**  \n   - **Rhetorical devices: Treat questions like "Wasn’t X agreed?" as assertions of X’s truth.**  \n   - Limit inferences: Base conclusions only on stated or logically derived information.  \n\nIllustrations:  \n- **Premise: "Should Lumina inquire, we’ll acknowledge Verdant is present."  \n  Hypothesis: "Verdant is present."  \n  → Entailment (conditional agreement).**  \n- **Premise: "Hadn’t I stated Azure is the meeting site?"  \n  Hypothesis: "Azure is where they convened."  \n  → Entailment (rhetorical confirmation).**  \n\nCategories: ...   | 92.9 |
> | Step 6 (Parent 1) | Determine ... by evaluating factual consistency, negation implications, and contextual alignment. \n\n Classification Rules  \n1. Contradiction: ... Explicit Factual Opposition: **Clear factual conflicts (e.g., "The road is dry" vs. "The road is wet")**.  \n   - Logical Incompatibility: Premise creates conditions that invalidate the hypothesis.  \n   - **Focus on factual clashes, not subjective disagreements (e.g., rejecting a belief ≠ rejecting the hypothesis)**. \n\n2. Entailment: ... Explicit Confirmation: Premise directly states or logically guarantees the hypothesis. \n   - **Contextual Support: Premise offers clear real-world validation (e.g., "He confirmed the event occurred")**.  \n   - **Definitions alone ≠ support unless explicitly tied to the hypothesis.** \n\n3. Neutral: ... **Non-Actionable Statements: Beliefs, assumptions, or emotions about the hypothesis ≠ proof (e.g., "I suspect X" ≠ "X is true")**.  \n   - Isolated Definitions: Explaining terms without applying them to the hypothesis**.  \n   - **Non-Committal Queries: Questions like "Do you think X?" ≠ factual claims**. \n\n Key Considerations: \n- Negation Implications:  \n  - **Premise negating intent (e.g., "He didn’t plan to go") ≠ contradiction of the action itself unless the action’s occurrence is denied.**  \n- Definitions vs. Assertions:  \n  - **Defining terms (e.g., "X refers to Y") ≠ entailment unless paired with a factual claim about the hypothesis.** | 91.1 |
>
> We highlight the case-specific additions in bold.
> In the first step, APO primarily adds general guidance with some specific details, resulting in a performance gain. However, later revisions tend to focus solely on narrowly tailored logic or memorized phrasings tied to individual examples, which fail to generalize and can even degrade performance.
>
> We hope these process illustrations will further clarify the optimization behaviors and help highlight how GRACE achieves more efficient and effective improvements.

---

### Note · Authors · 2025-08-12

We express our sincere gratitude to all reviewers for their thoughtful and constructive feedback!

We are glad that multiple reviewers acknowledged the importance of automatic prompt optimization and found our GRACE method well-motivated, clearly presented, and empirically validated. In particular:
**Principled and effective design**: GRACE addresses instability in prior methods through gated refinement and adaptive compression, balancing local and global updates for stable, efficient optimization (R1–R4);
**Clarity and empirical rigor**: Reviewers praised our clear exposition, step-by-step examples, and ablation studies, supported by strong results across diverse tasks and models (R1, R2, R3).
**High performance–cost ratio**: GRACE achieves better results while reducing optimization cost by up to 75%, making it practical and scalable (R1, R3, R4).

We have carefully addressed each reviewer’s concerns to improve clarity, completeness, and applicability of our work:

R1 asked whether the number of search steps and cost directly reflect computing time, and suggested including step-by-step optimization examples for baselines.
We have clarified the relation between search size, compute time, and cost, with decomposed analysis and estimations. We also included detailed optimization traces for baseline methods like OPRO and APO to aid comparison.

R2 requested more guidance on hyperparameter selection and a clearer discussion of limitations.
We have expanded ablation studies and offer practical default settings for new tasks. Additionally, we further discussed challenges on domain-specific tasks and propose retrieval-based extensions as a future direction.

R3 raised concerns about sensitivity to meta-prompt design and generalizability on more tasks.
We have tested paraphrased meta-prompts and found GRACE to be robust, and added new summarization benchmarks to demonstrate broader applicability.

R4 suggested exploring the transferability of optimized prompts and the impact of different optimizer LLMs.
We have evaluated prompt transfer across base LLMs and experimented with different optimizer models. GRACE remains consistently effective, achieving top performance across all configurations.

Finally, we sincerely thank the reviewers and AC for their time and insights.
We are excited to contribute toward more robust, interpretable, and efficient prompt optimization, and we hope our work will inspire continued progress in this direction.

---

### Decision · Program_Chairs · 2025-09-17

**Decision:**

Accept (poster)

**Comment:**

The paper introduces a new prompt optimization method that solves two key challenges: instability in prompt update (by a gated refinement mechanism), and tendency to stuck in local optima (by a novel adaptive compressive technique).

This is a timely work studying an important problem. The motivations are clear. The proposed methods are novel and effective. Experimental results show large and consistent improvements across different datasets and tasks.

All reviewers recommended acceptance, albeit with varying degrees of support. The following major concerns from the reviewers who rated the work borderline accept (4) were well addressed by the authors using additional experiments.
 scUS
* sensitivity to meta prompts.
* tasks are limited to classification and reasoning.
NhAd
* comparative experiments to verify the effectiveness and scalability of the proposed method.

Overall, the AC finds that this is a solid work addressing an important problem. Please add the additional experiments and clarifications in the final version.